# An evolutionary recent IFN/IL-6/CEBP axis is linked to monocyte expansion and tuberculosis severity in humans

Murilo Delgobo[1], Daniel AGB Mendes[1], Edgar Kozlova[1],
Edroaldo Lummertz Rocha[1,2], Gabriela F Rodrigues-Luiz[1], Lucas Mascarin[1],
Greicy Dias[1], Daniel O Patrício[1], Tim Dierckx[3], Maíra A Bicca[1], Gaëlle Bretton[4],
Yonne Karoline Tenório de Menezes[1], Márick R Starick[1], Darcita Rovaris[5],
Joanita Del Moral[6], Daniel S Mansur[1], Johan Van Weyenbergh[3]*, André Báfica[1]*

[1]Laboratório de Imunobiologia, Departamento de Microbiologia, Imunologia e Parasitologia, Universidade Federal de Santa Catarina, Florianópolis, Brazil; [2]Boston Children's Hospital, Boston, United States; [3]Department of Microbiology, Immunology and Transplantation, Rega Institute for Medical Research, Laboratory for Clinical and Epidemiological Virology, KU Leuven, Leuven, Belgium; [4]Laboratory of Molecular Immunology, The Rockefeller University, New York, United States; [5]Laboratório Central do Estado de Santa Catarina/LACEN, Florianópolis, Brazil; [6]Serviço de Hematologia, Hospital Universitário, Universidade Federal de Santa Catarina, Florianópolis, Brazil

*For correspondence:
j.vw@live.be (JVW);
andre.bafica@ufsc.br (AB)

Competing interests: The authors declare that no competing interests exist.

**Abstract** Monocyte counts are increased during human tuberculosis (TB) but it has not been determined whether *Mycobacterium tuberculosis* (*Mtb*) directly regulates myeloid commitment. We demonstrated that exposure to *Mtb* directs primary human CD34+ cells to differentiate into monocytes/macrophages. In vitro myeloid conversion did not require type I or type II IFN signaling. In contrast, *Mtb* enhanced IL-6 responses by CD34+ cell cultures and IL-6R neutralization inhibited myeloid differentiation and decreased mycobacterial growth in vitro. Integrated systems biology analysis of transcriptomic, proteomic and genomic data of large data sets of healthy controls and TB patients established the existence of a myeloid *IL-6/IL6R/CEBP* gene module associated with disease severity. Furthermore, genetic and functional analysis revealed the *IL6/IL6R/CEBP* gene module has undergone recent evolutionary selection, including Neanderthal introgression and human pathogen adaptation, connected to systemic monocyte counts. These results suggest *Mtb* co-opts an evolutionary recent IFN-IL6-CEBP feed-forward loop, increasing myeloid differentiation linked to severe TB in humans.
DOI: https://doi.org/10.7554/eLife.47013.001

## Introduction

Hematopoiesis, the development of different blood cell lineages from hematopoietic stem cells (HSCs), is a fundamental physiological process in vertebrates. HSCs give rise to lineage-restricted progenitors that gradually differentiate into mature cells. Following cellular differentiation, single-lineage elements including erythrocytes, megakaryocytes, lymphocytes as well as myeloid cells such as monocytes and granulocytes circulate throughout the body performing diverse functions. While HSC development towards cellular lineages during homeostasis has been extensively studied (*Hoggatt et al., 2016*), the mechanisms by which how progenitors give rise to mature cells during stress responses are less comprehended. For instance, certain pathogens regulate production of

blood cells by the bone marrow and it has been shown that fine-tuned regulation of cytokine-induced signals is required for differentiation of HSC into mature cell types (*Kleppe et al., 2017*; *Mirantes et al., 2014*; *Zhang and Lodish, 2008*). For example, the protozoan parasite that causes kalazar, *Leishmania donovani,* inhabits the bone marrow of humans (*Kumar and Nylén, 2012*), targets bone marrow stromal macrophages (*Cotterell et al., 2000*) and induces differentiation of myeloid cells at the expense of lymphoid progenitors (*Abidin et al., 2017*; *Cotterell et al., 2000*). In the same line of evidence, after experimental exposure to Gram-negative bacteria, mice display increased amounts of bone marrow-derived neutrophils, through a G-CSF–C/EBPα dependent mechanism (*Boettcher et al., 2014*). Moreover, infection by intracellular bacteria has been shown to modulate production of circulating leukocytes involving IFN-γ-mediated pathways (*Baldridge et al., 2010*; *MacNamara et al., 2011*; *Murray et al., 1998*). Altogether, these studies indicate vertebrate hosts respond to infection by 'remodeling' cell lineage production, which are highly dependent upon the interplay of cytokine-induced hematopoiesis triggered during infection. Interestingly, recent reports have demonstrated hematopoietic stem/progenitor cells (HSPCs) may be infected by different classes of infectious agents such as viruses and bacteria, albeit at low efficiency (*Carter et al., 2011*; *Kolb-Mäurer et al., 2002*). Therefore, since many pathogens may reach the bone marrow and provide microbial-HSC interactions, it is possible that, in addition to cytokines, pathogen recognition by progenitor cells directly regulate cell lineage commitment providing an anti-microbial defense system. In contrast, the Red Queen hypothesis (*Van Valen, 1973*) predicts such pathogens would benefit from cell lineage commitment to establish themselves into the host.

The human pathogen *Mycobacterium tuberculosis* (*Mtb*) has been recently detected in circulating HSCs (Lin⁻CD34⁺) from latent TB individuals (*Tornack et al., 2017*). Since *Mtb* can also gain access to the bone marrow during extra-pulmonary (*Mert et al., 2001*) as well as active pulmonary TB (*Das et al., 2013*), it has been suggested that the human bone marrow is a niche/reservoir for this bacterium during natural pathogen infection. However, whether interactions between *Mtb* and human CD34⁺ cells drive cellular differentiation has not been formally demonstrated. Interestingly, earlier (*Rogers, 1928*; *Schmitt et al., 1977*) and recent (*Berry et al., 2010*; *Zak et al., 2016*) studies have reported major changes in the peripheral myeloid cells such as increased blood counts and dysregulated 'interferon transcriptional signature' during active TB. More specifically, several 'interferon-stimulated genes' (ISGs) are modulated in circulating mature neutrophils and monocytes in active TB patients, which calls forth a possible role of such genes in TB pathogenesis (*Berry et al., 2010*; *Dos Santos et al., 2018*; *Zak et al., 2016*). In contrast, lymphocyte compartments were recently demonstrated to be contracted during progression from latent to active TB in humans (*Scriba et al., 2017*). Therefore, the observed changes in blood leukocytes could be a consequence of the interactions between Mtb and the bone marrow cellular environment. Thus, we hypothesized that *Mtb* regulates cellular differentiation of human HSPCs. By employing in vitro functional assays and integrated systems biology analysis of published available cohorts of healthy controls and TB patients, our study suggests that *Mtb* co-opts an evolutionarily recent IFN/IL-6/CEBP axis linking monocyte differentiation and disease severity.

## Results

### *Mtb* H37Rv replicates in primary human CD34⁺ cell cultures

To investigate the dynamics of *Mtb* infection by HSPCs, we have exposed peripheral blood mononuclear cells (PBMCs) from healthy donors to H37Rv *Mtb* (multiplicity of infection, MOI3) and measured bacterial infectivity by CD34⁺ cells. First, by using a fluorescent dye (syto-24) which does not influence bacteria infectivity (data not shown), flow cytometry experiments demonstrated that *Mtb* were associated with CD34⁺ cells following 4 hr exposure to mycobacteria (*Figure 1a,b* and *Figure 1—figure supplement 1a*). At that time point, we observed ~69% of CD34⁺ and ~79% of CD14⁺ associated with *Mtb* (*Figure 1c*). When compared to CD14⁺ cells, which are highly phagocytic cells, the MFI measurements within the CD34⁺ cell population were found to be ~4 x lower (*Figure 1d*). These data suggest that PBMC CD34⁺ cells may be permissive to *Mtb* infection in vitro. Similarly, purified cord blood derived CD34⁺ cells display comparable % and MFI as those seen in PBMC CD34⁺ cells (*Figure 1—figure supplement 1b*). Confocal microscopy analysis confirmed the presence of sparse intracellular mycobacteria in purified cord blood derived CD34⁺ cells at 4 hr post-infection (pi)

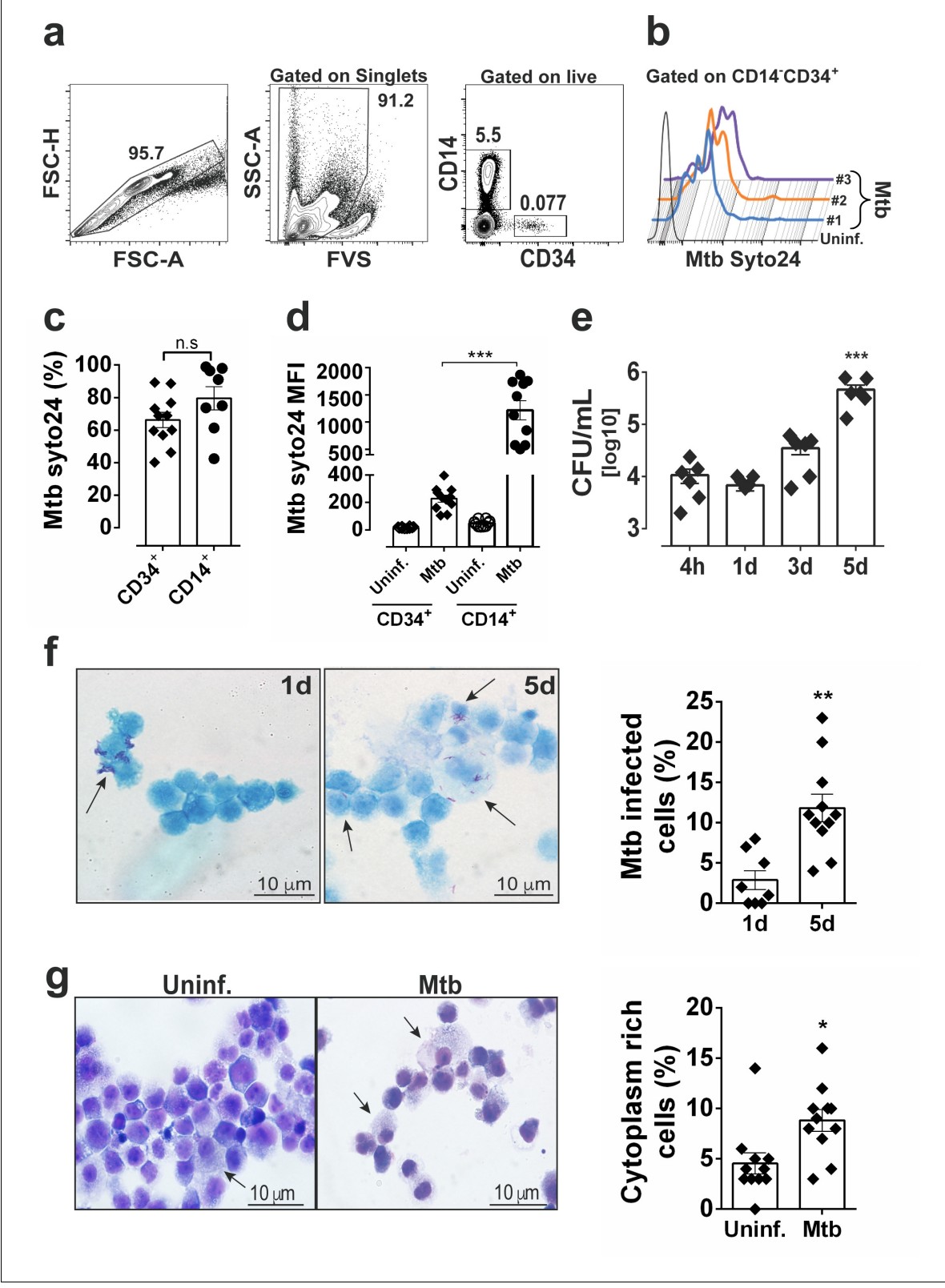

**Figure 1.** *Mtb* H37Rv infects human CD34+ cells and proliferates in cell cultures in vitro. PBMC from healthy donors were exposed to syto24-labeled *Mtb* H37Rv (MOI3, *Figure 1—figure supplement 1a*) for 4 hr. (a) Representative flow cytometry contour plots of gating strategy to analyze *Mtb* syto24 association in FVS-negative (live) CD34+ events and CD14+ events. (b) Live CD34+Lin− events gated in a were analyzed for *Mtb*-Syto24 MFI. Black line: Uninfected control. Blue, orange and purple lines represent samples from three different donors. (c) Frequencies and (d) MFI of *Mtb* syto24+ events in

*Figure 1 continued on next page*

*Figure 1 continued*

CD34[+] or CD14[+] events gates from uninfected or *Mtb* syto24-exposed bulk PBMCs. Results are means ± SEM of data pooled from three independent experiments, n = 10 healthy donors. ***p≤0.001 between *Mtb* syto24 CD34[+] vs CD14[+] groups. (e,) Purified cord blood-derived CD34[+] cells were exposed to *Mtb* H37Rv (MOI3) for different time points and CFUs from cell culture lysates were enumerated in 7H10 media. Results are means ± SEM of data pooled from five independent experiments, ***p≤0.001 between 5d vs 4 hr groups. (f) Kinyoun staining of CD34[+] cells after 1d and 5d of infection and quantification, as described in the methodology section, shown in the right panel. Arrows indicate cells associated with bacilli. Experiments shown are representative of two performed. **p≤0.01 between 5d vs 1d groups. (g) Representative Giemsa staining of CD34[+] cells of 5d-cultures and quantification, as described in the methodology section, shown in the right panel. Arrow indicates cytoplasm-rich cells in *Mtb*-infected cultures and uninfected cultures. Experiments shown are representative of two performed. *p≤0.05 between *Mtb* vs uninfected groups.

DOI: https://doi.org/10.7554/eLife.47013.002

The following source data and figure supplements are available for figure 1:

**Source data 1.** Raw data from *Figure 1*.
DOI: https://doi.org/10.7554/eLife.47013.005

**Figure supplement 1.** *Mtb*-CD34[+] interactions and signaling pathways associated with HSPC differentiation.
DOI: https://doi.org/10.7554/eLife.47013.003

**Figure supplement 1—source data 1.** Raw data from *Figure 1—figure supplement 1*.
DOI: https://doi.org/10.7554/eLife.47013.004

(*Figure 1—figure supplement 1c*). These findings raise the possibility that although human primary Lin[-]CD34[+] cells can be infected by *Mtb* in vitro and in vivo (*Tornack et al., 2017*), this cell population may display intrinsic resistance to *Mtb* infection as it has been reported for other bacterial species (*Kolb-Mäurer et al., 2002*). Next, we employed a purified cell culture system to investigate whether H37RV *Mtb* replicates in CD34[+] cells in different time points. When sorted purified cord-blood CD34[+] cells were exposed to *Mtb* H37Rv (MOI3) and cultivated in StemSpan SFEM II (*Bodine et al., 1991*; *Keller et al., 1995*), bacilli numbers exhibited a ~ 1.5 log growth at 5 days post infection (dpi) (*Figure 1e*). As a control, purified CD14[+] cells displayed higher bacterial proliferation than purified CD34[+] cells over time (*Figure 1-Figure 1—figure supplement 1d*). Together, these findings demonstrate that *Mtb* infects and replicates in primary human CD34[+] cell cultures in vitro. While at one dpi bacilli were more associated with the surface of round cells (*Figure 1f*), at five dpi intracellular bacteria were associated with cells with abundant cytoplasm (*Figure 1f*), suggesting that *Mtb*-exposed cultures displayed increased frequencies of cells exhibiting morphological alterations over time (*Figure 1f*, right panel). Indeed, Giemsa staining (*Figure 1g*, arrows) presented higher frequency of cytoplasm- richer cells in bacteria-exposed vs uninfected cell cultures (*Figure 1g*, right panel), thus suggesting that *Mtb* infection enhances cellular differentiation by CD34[+] cells in vitro.

## Live *Mtb* induces CD34[+] cells towards myeloid differentiation and monocyte output

Next, to investigate whether *Mtb* triggered cellular differentiation by human CD34[+] cells, we evaluated differential expression of 180 transcription factors (TFs) associated with differentiation of distinct hematopoietic cells (*Novershtern et al., 2011*) in RNA-seq samples of *Mtb*-exposed purified cells (*Figure 1—figure supplement 1e – Figure 2—source data 1*). Interestingly, *Mtb* infection increased the expression of lineage-specific regulators of myeloid (GRAN/MONO) (*SPI1, CEBPB, CEBPA, EGR2 and STAT2*), but not lymphoid (B and T CELL) (*GABPA, SOX5, TCF3, GATA3, LEF1, RORA and LMO7*) or megakaryoid/erythroid (EARLY/LATE ERY) (*GATA1, FOXO3, NFE2, TAL1*) differentiation (*Figure 2a – Figure 2—source data 1*). Similarly, CellRouter (*Lummertz da Rocha et al., 2018*) signature score of the GRAN/MONO gene set was found to be increased in *Mtb* vs uninfected samples in all time points studied (*Figure 2b - Figure 2—source data 1*). Next, we applied a network biology-built computational platform (*Cahan et al., 2014*) which, based on classification scores, can assess the extent to which a given population resembles mature cell types. *Figure 2c* (*Figure 2—source data 1*) shows that mRNA samples from *Mtb*-exposed CD34[+] cells presented enrichment of monocyte/macrophage profiles, but not other mature cell populations such as lymphocytes or dendritic cells. Additionally, at five dpi, CD34[+] cell cultures displayed increased frequencies of cells positive for CD11b (*Figure 2d*), a surface molecule expressed during myeloid differentiation (*Hickstein et al., 1989*; *Rosmarin et al., 1989*). Together, these data suggest that *Mtb* drives human primary CD34[+] cells towards myeloid differentiation. We next employed flow

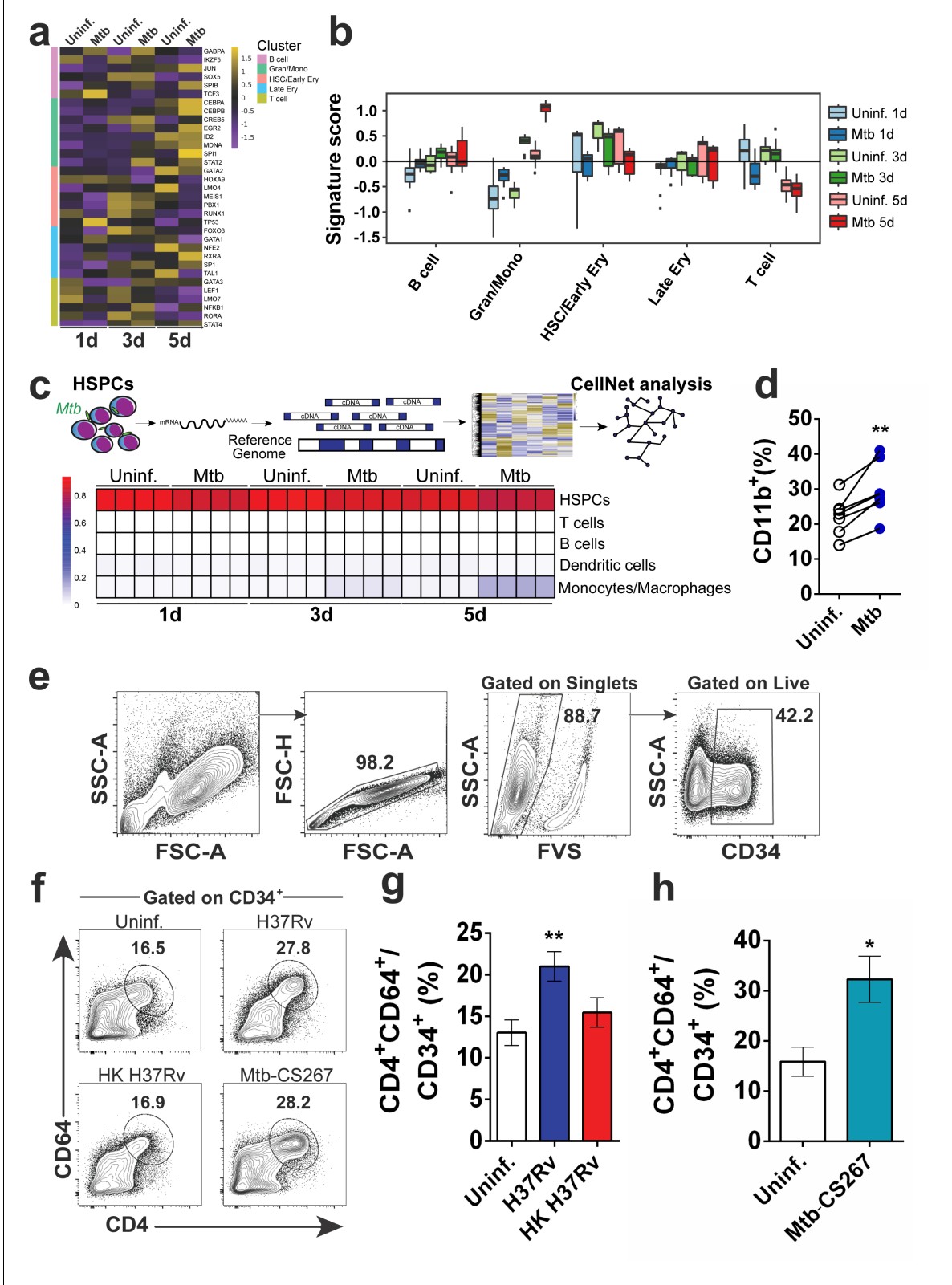

**Figure 2.** Live *Mtb* induces human CD34[+] cells towards myeloid differentiation in vitro. Purified CD34[+] cells from healthy donors (n = 3) were exposed to *Mtb* H37Rv (MOI3) for different time points and mRNA-seq was performed as described in methodology section. (a) Heatmap of the mRNA expression (z-score) of transcription factors involved in cell lineage commitment (*Novershtern et al., 2011*). (b) Signature score of data from a) by employing CellRouter analysis. (c) Heatmap from mRNA data of uninfected vs *Mtb* infected cultures analyzed by CellNet. (d) Purified CD34[+] cells were
*Figure 2 continued on next page*

*Figure 2 continued*

exposed to *Mtb* H37Rv (MOI3) for 5 days and flow cytometry was performed. Graph represents frequencies of CD11b+ events in uninfected (open circles) vs *Mtb*-infected cultures (blue circles) from four independent experiments. **p≤0.01 between *Mtb* and uninfected groups. (e) Purified CD34+ cells were exposed to *Mtb* H37Rv, Heat-killed (HK) *Mtb* H37Rv or *Mtb* clinical isolate 267 (*Mtb*-CS267) (MOI3) for 5 days and flow cytometry with the gating strategy was performed. (f) Representative contour plots show frequencies of CD4+CD64+ events in CD34+ events. CD34+CD4+CD64+ events of polled data from f) were plotted to generate bar graphs (g) and (h). Results are means ± SEM of data pooled from four independent experiments (g) and two independent experiments (h). (g) ** indicates p≤0.01 between H37Rv vs uninfected or HK H37RV groups. (h) * indicates p≤0.05 between *Mtb*-CS267 vs uninfected groups.

DOI: https://doi.org/10.7554/eLife.47013.006

The following source data and figure supplements are available for figure 2:

**Source data 1.** Raw data from *Figure 2*.

DOI: https://doi.org/10.7554/eLife.47013.009

**Source data 2.** Counts matrix of RNAseq data of *Mtb*-exposed and control CD34+ cell transcriptomes.

DOI: https://doi.org/10.7554/eLife.47013.010

**Figure supplement 1.** Myeloid differentiation by PBMC or bone marrow CD34+ cells exposed to *Mtb,* mycobacterial ligands and cell death analysis in vitro.

DOI: https://doi.org/10.7554/eLife.47013.007

**Figure supplement 1—source data 1.** Raw data from *Figure 2—figure supplement 1*.

DOI: https://doi.org/10.7554/eLife.47013.008

cytometry to measure cell surface molecules previously associated with myeloid differentiation of human CD34+ cells (*Cimato et al., 2016*; *Gorczyca et al., 2011*; *Kawamura et al., 2017*; *Manz et al., 2002*; *Olweus et al., 1995*). More specifically, we gated on CD34+ events and quantified % CD64+CD4+ cells. Corroborating our hypothesis, *Mtb* enhanced the frequency of CD64+CD4+CD34+ cells (*Figure 2e–h*) at five dpi, but not CD10+CD34+ (lymphoid) or CD41a+CD34+ (megakaryoid) progenitors (*Figure 2—figure supplement 1a*). Similarly, Lin-CD34+ cells from PBMC (*Figure 2—figure supplement 1b*) samples from healthy donors exposed to *Mtb* displayed increased frequency of CD4+CD64+CD34+ cells and augmented levels of CD38 and HLA-DR (*Figure 2—figure supplement 1b,c*), two molecules associated with advanced stage of cellular differentiation (*Cimato et al., 2016*; *De Bruyn et al., 1995*; *Terstappen et al., 1991*). When compared to their base line levels, we also observed an increased % CD4+CD64+CD34+ cells in *Mtb*-exposed bulk bone marrow samples from two healthy individuals (*Figure 2—figure supplement 1c*, p=0.08). In addition, frequencies of CD4+CD64+CD34+ cells were higher in cultures infected to live H37Rv *Mtb* than those exposed to heat-killed (HK) bacteria (*Figure 2f,g*). These results suggest that the observed cellular phenotype was mostly due to the activities of live pathogen and only partially to mycobacterial PAMPs such as TLR2 (Ara-LAM) or TLR9 (*Mtb* gDNA) agonists (*Bafica et al., 2005*; *Underhill et al., 1999*), which induced CD38 and HLA-DR, but not CD4 and CD64 expression in CD34+ cells (*Figure 2—figure supplement 1d*). Importantly, increased frequency of CD4+CD64+CD34+ cells was also observed when cell cultures were exposed to a clinical isolate of *Mtb* (*Figure 2h*), ruling out a possible genetic factor associated with the laboratory strain H37Rv (*Brites and Gagneux, 2015*). Furthermore, CD34+ cell death was not enhanced by *Mtb* infection as demonstrated by the use of a live-and-dead probe and lactate dehydrogenase (LDH) quantification in cell culture supernatants (*Figure 2—figure supplement 1e,f*). Together, these data indicate live *Mtb* directs primary human CD34+ cells towards myeloid differentiation in vitro.

Next, to investigate whether *Mtb* enhanced HSPC differentiation into mature myeloid populations (*Kawamura et al., 2017*; *Lee et al., 2015*; *Manz et al., 2002*), purified CD34+ cells were exposed to *Mtb* and 10d later, surface molecules were measured by flow cytometry. Live H37Rv (*Figure 3a,e*) or clinical isolate *Mtb*, (*Figure 3—figure supplement 1a*) but not *Leishmania infantum* promastigotes (data not shown), enhanced expression of the monocyte surface molecule CD14, confirming the observed monocyte/macrophage output enrichment by CellNet analysis (*Figure 2c*). Interestingly, in this in vitro culture system, CD14+ cells started to emerge at low levels at day three in both uninfected and *Mtb*-exposed cell cultures, while enhancement in monocyte frequency was observed in *Mtb*-stimulated cells at later time points, that is at 7 and 10 dpi. (*Figure 3—figure supplement 1b*). Compared to control cell cultures, CD14+ cells induced by *Mtb* displayed similar MFI expression of CD11b, HLA-DR, CD64 and CD16 surface molecules (*Figure 3i*) and most, but not all

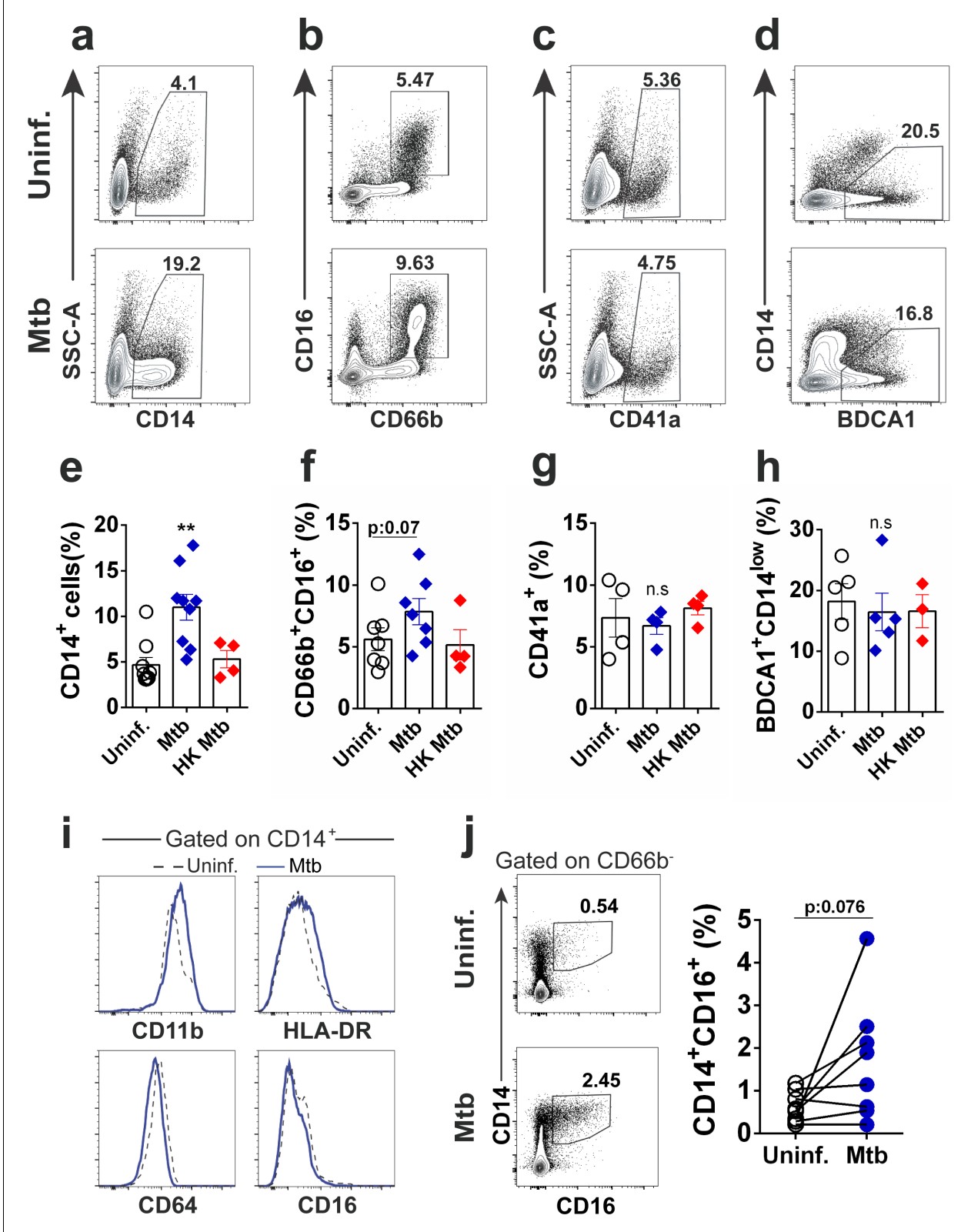

**Figure 3.** *Mtb* infection increases monocyte output from CD34[+] cells in vitro. Purified CD34[+] cells were exposed to live *Mtb* H37Rv or HK *Mtb* H37Rv (MOI3) for 10 days and flow cytometry was employed to determine the mature cell frequencies in the cell cultures. Representative dot plots of (a) monocytes (CD14[+]), (b) neutrophils (CD16[+]CD66b[+]), (c) megakaryocytes/platelets (CD41a[+]) and (d) classical myeloid dendritic cells (BDCA1[+]CD14[low]) in uninfected and *Mtb*-infected CD34[+] cell cultures. Graphs show frequencies of (e) CD14[+], (f) CD16[+]CD66b[+], (g) CD41a[+] and (h) BDCA1[+]CD14[low] events

*Figure 3 continued on next page*

Figure 3 continued

in uninfected (open circles), live *Mtb*-infected (blue diamonds) or HK *Mtb*-exposed (red diamonds) cell cultures at day 10. Each symbol represents one individual experiment. Results are means ± SEM of data pooled from 3 to 9 independent experiments. **p≤0.01 between *Mtb* vs uninfected or HK *Mtb* groups. (i,) Histograms show the expression of CD11b, HLA-DR, CD64 and CD16 in CD14$^+$ events from a). Black dashed lines: Uninfected control. Blue solid lines: *Mtb*-infected group. Data representative of 5 independent experiments. (j,) Frequency of CD14$^+$CD16$^+$ events in *Mtb*-exposed cell cultures after 10d. Contour dot plot of CD14$^+$CD16$^+$ frequencies from one representative donor. Open circles: Uninfected control. Blue circles: *Mtb*-infected group. Each symbol represents an individual experiment. Pooled data of eight independent experiments, n = 5 different donors. p=0.076 between *Mtb* vs uninfected groups.

DOI: https://doi.org/10.7554/eLife.47013.011

The following source data and figure supplements are available for figure 3:

**Source data 1.** Raw data from *Figure 3*.

DOI: https://doi.org/10.7554/eLife.47013.014

**Figure supplement 1.** Monocyte differentiation and reactome pathways associated to *Mtb*-exposed CD34+ cells in vitro.

DOI: https://doi.org/10.7554/eLife.47013.012

**Figure supplement 1—source data 1.** Raw data from *Figure 3—figure supplement 1*.

DOI: https://doi.org/10.7554/eLife.47013.013

experiments presented increased frequency of CD14$^+$CD16$^+$ monocytes (*Figure 3j*), which were previously associated with severe pulmonary TB (*Balboa et al., 2011*). Moreover, albeit not statistically significant, *Mtb* enhanced the frequency of CD16$^+$CD66b$^+$ neutrophils in the majority but not all samples tested (*Figure 3b,f*). In contrast, HK *Mtb* did not stimulate monocyte or neutrophil output (*Figure 3e,f*). As expected (*Figure 2—figure supplement 1a*), megakaryoid/platelet- (*Figure 3c,g*), dendritic cell- (*Figure 3d,h*) or erythroid- (*Figure 3—figure supplement 1c*) associated markers were unchanged after exposure to live or HK *Mtb*. Altogether, these results suggest *Mtb* selectively favors the generation of monocytes and, to a lesser extent, neutrophils, by human CD34$^+$ cells in vitro.

## In vitro *Mtb*-enhanced myeloid differentiation is mediated by IL-6R, but not type I or type II IFN signaling

Cytokines are important triggers of Lin$^-$CD34$^+$ differentiation in vivo and in vitro (*Endele et al., 2014*; *Hoggatt et al., 2016*; *Zhang and Lodish, 2008*) and Reactome pathway analysis of genes differentially expressed between *Mtb*-infected versus uninfected conditions displayed enrichment of 'cytokine signaling in immune system' (*Figure 3—figure supplement 1d*; n = 3 donors, two independent experiments - *Figure 2—source data 1*). Among several genes, we observed a significant enrichment of *IL6* (*Supplementary file 1 - Figure 2—source data 1*), a key HSPC-derived regulator of myeloid differentiation in mouse and human models (*Jansen et al., 1992*; *Zhao et al., 2014*) which was confirmed in our system by the addition of exogenous IL-6 to CD34$^+$ cells (*Figure 4—figure supplement 1a*). Moreover, cytokine receptors, including *IL6R*, as well as their cytokine partners, containing *IL6*, were enriched in *Mtb*-exposed CD34$^+$ cell cultures (*Figure 4a,b* and *Figure 4—figure supplement 1b - Figure 2—source data 1*). Similarly, increased *IL6* expression was confirmed by qPCR (*Figure 4—figure supplement 1c*). In addition, 'interferon signaling' and 'interferon alpha/beta' pathways were significantly enriched in *Mtb*-exposed CD34$^+$ cells (*Figure 3—figure supplement 1d* and *Supplementary file 1 - Figure 2—source data 1*). This was confirmed in five donors, which displayed increased levels of *IFNA2*, *IFNB* and *IFNG* transcripts, albeit at a lower level relative to *IL6* mRNA (*Figure 4—figure supplement 1c*). Importantly, interferon-stimulated genes (ISGs) such as *MX1*, *ISG15* and *IFI16* as well as IL-6R-stimulated genes such as *IL1RA*, *GRB2* and *CXCL8* were enhanced in *Mtb*-stimulated CD34$^+$ cells from five different donors (*Figure 4—figure supplement 1d*), suggesting IL-6 and IFN signaling are active in these cells. Corroborating previous findings showing that HSPCs produce IL-6 following microbial stimuli (*Allakhverdi and Delespesse, 2012*), live *Mtb* also induced intracellular IL-6 production in Lin$^-$CD34$^+$ cells from bacteria-exposed PBMC 1d cultures (*Figure 4c*). This was confirmed in 1d culture supernatants of purified CD34$^+$ cells exposed to live *Mtb* which presented augmented levels of IL-6, but not IFN-γ, IL-1β or TNF (*Figure 4—figure supplement 1e*). Interestingly, while HK *Mtb* also stimulated production of IL-6 (*Figure 4c*), dead bacteria did not induce CD38, CD4 and CD64 expression in PBMC Lin-CD34$^+$ cells as seen in cell cultures exposed to live *Mtb* (*Figure 4—figure supplement 1f*). When compared to

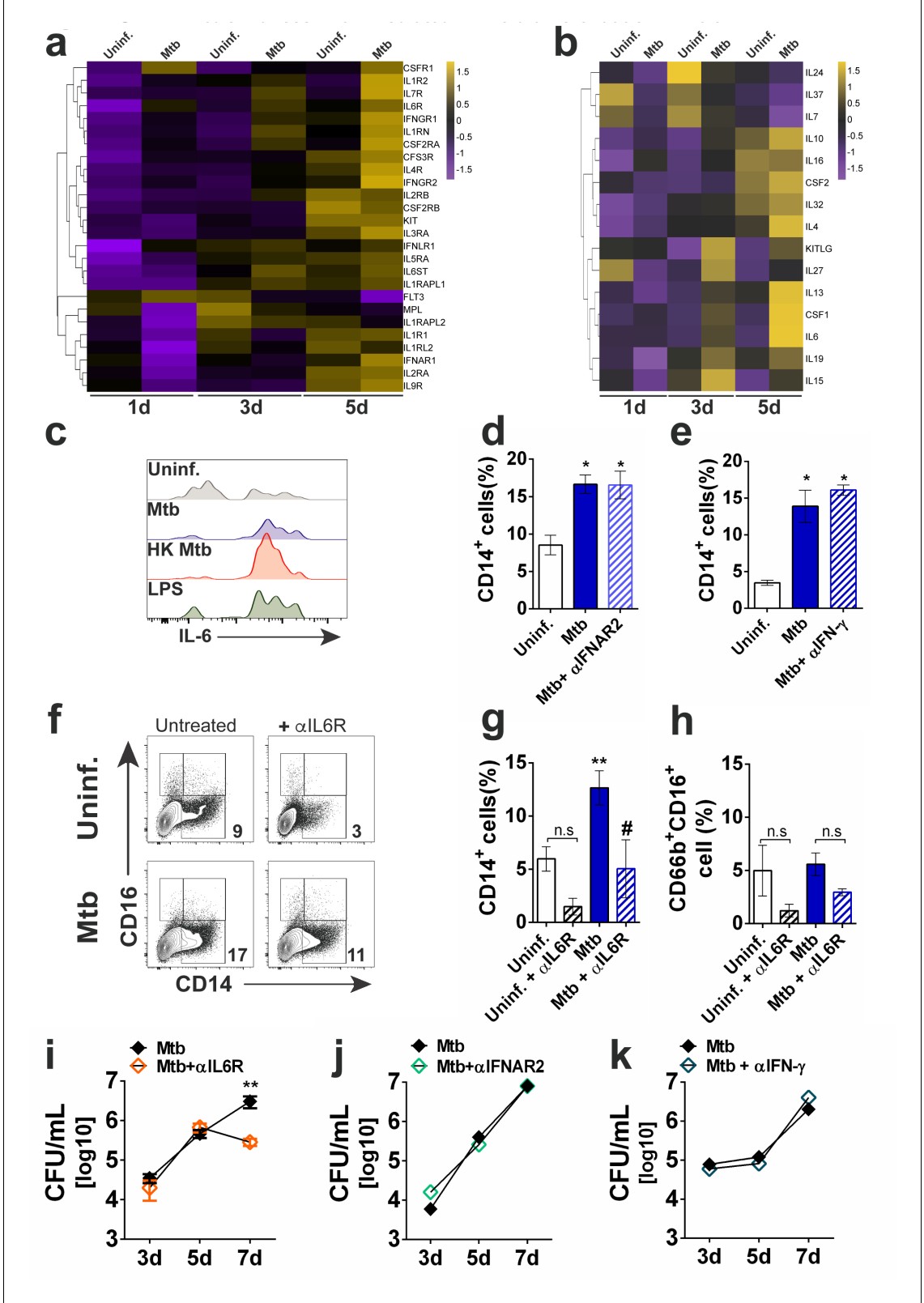

**Figure 4.** *Mtb* enhances IL-6R-mediated myeloid differentiation in vitro. Purified CD34[+] cells were exposed to *Mtb* H37Rv (MOI3) for different time points and mRNA-seq was performed as described in the methodology section. (a) Heatmap (z-score) of differentially expressed cytokine receptor genes. (b) Heatmap (z-score) of differentially expressed cytokine genes. Shown is the average mRNA expression of three different donors from two independent experiments. (c) PBMC from healthy donors were exposed to Mtb H37Rv, HK Mtb or LPS (100 ng/mL) for 24 hr and intracellular IL-6 was

*Figure 4 continued on next page*

*Figure 4 continued*

detected by flow cytometry. Live CD34+Lin- events gated as in *Figure 1a* were analyzed for IL-6 MFI. Representative histogram from two independent experiments. Purified CD34$^+$ cells were treated with (d) α-IFNAR2 (1 μg/ml) or (e) α-IFN-γ (10 μg/ml) and then exposed to *Mtb* H37Rv (MOI3) during 10d for determination of CD14$^+$ monocyte frequencies. Results are means ± SEM of data pooled from two independent experiments. *p≤0.05 between *Mtb*, α-IFNAR2 or α-IFN-γ vs uninfected groups. (f) Representative contour plots of CD14$^+$ monocytes in CD34$^+$ cell cultures exposed to *Mtb*, in the presence or absence of α-IL6R (Tocilizumab, 1 μg/ml) for 10d. (g) Results shown are means ± SEM of data pooled from three independent experiments from (f) **p≤0.01 between *Mtb* vs uninfected groups and #p≤0.05 between *Mtb* and *Mtb*+α-IL6R-treated groups. (h) Results shown are means ± SEM of data pooled from three independent experiments showing frequency of CD66$^+$CD16$^+$ neutrophils in *Mtb*-infected cell cultures in the presence or absence of α-IL6R. Purified CD34$^+$ cell cultures were treated as in (d–f) with (i) α-IL6R, (j) α -IFNAR2 and (k) α-IFN-γ and then exposed to *Mtb* (MOI3) for different time points and CFU enumerated as described in the methodology section. Results are means ± SEM of data pooled from four independent experiments. **p≤0.01 between *Mtb* and *Mtb*+ α-IL6R at 7d.
DOI: https://doi.org/10.7554/eLife.47013.015

The following source data and figure supplements are available for figure 4:

**Source data 1.** Raw data from *Figure 4*.
DOI: https://doi.org/10.7554/eLife.47013.018

**Figure supplement 1.** Gene expression and cytokine production during myeloid differentiation in vitro.
DOI: https://doi.org/10.7554/eLife.47013.016

**Figure supplement 1—source data 1.** Raw data from *Figure 4—figure supplement 1*.
DOI: https://doi.org/10.7554/eLife.47013.017

the live pathogen, qPCR experiments with HK *Mtb*-exposed purified CD34$^+$ cells did not show induction of ISG *STAT1* (*Figure 4—figure supplement 1g*), suggesting the existence of cross talking regulatory pathways between live *Mtb*, IL-6 and IFN signaling to boost myeloid differentiation in vitro. Since these data pointed that IL-6 and IFN signaling are potential pathways involved in *Mtb*-enhanced myeloid differentiation by CD34$^+$ cells, we employed neutralizing monoclonal antibodies as a tool to investigate this possibility. While type I IFN signaling was necessary for *Mtb*-stimulated ISGs such as *STAT1* and *MX1* transcription (*Figure 4—figure supplement 1h*), neither type I nor type II IFN signaling pathways were required for *Mtb*-enhanced monocyte/granulocyte conversion (*Figure 4d,e* and *Figure 4—figure supplement 1i*). In contrast, neutralizing anti-IL-6Ra antibody (α-IL6R) inhibited background levels of CD14$^+$ monocytes and CD66b$^+$ granulocytes, as well as *Mtb*-enhanced myeloid differentiation by CD34$^+$ cell cultures (*Figure 4f–h*) but not transcription of *STAT1* and *MX1* (*Figure 4—figure supplement 1h*). In addition, megakaryoid, erythroid- or dendritic cell-associated surface molecules were unaltered in α-IL6R-treated cell cultures (*Figure 4—figure supplement 1j–l*). Interestingly, *Mtb*-exposed CD34$^+$ cell cultures treated with α-IL6R (*Figure 4i*) presented significantly lower CFU counts when compared with infected untreated control cell cultures, while α-IFNAR2 (*Figure 4j*) or α-IFN-γ (*Figure 4k*) did not affect CFU counts. Together, these results suggest live *Mtb* enhances IL-6R-mediated myeloid differentiation by human CD34$^+$ cells in vitro.

## An *IL6/IL6R/CEBPB* gene module is enriched in the active TB transcriptome and proteome

To investigate whether IL-6R signaling correlates with monocyte expansion and TB-associated pathology in vivo, we performed a comprehensive systems biology analysis integrating several large transcriptomic and proteomic data sets from published cohorts of healthy controls and patients with latent, active and disseminated TB (*Berry et al., 2010*; *Hecker et al., 2013*; *Naranbhai et al., 2015*; *Novikov et al., 2011*; *Scriba et al., 2017*) (*Supplementary file 2*). First, we used Ingenuity Pathway analysis (IPA) to determine IL-6/IL-6R upstream regulators in transcriptomes from publicly available CD14$^+$ monocytes of active TB patients (*Berry et al., 2010*).. As shown in *Figure 5a* (top panel), *IL6*, *IL6ST*, *IL6R* and *STAT3* were significantly enriched in transcriptomes of active TB monocytes, when compared to cells from healthy controls. As reported previously (*Berry et al., 2010*; *Mayer-Barber et al., 2011*; *Novikov et al., 2011*), *STAT1* and *IL1B* were also confirmed as upstream regulators in active TB monocytes (*Figure 5a*, top panel). We next examined potential genes share between IL6/IL6R and type I IFN signaling pathways in active TB monocytes. Strikingly, the two top upstream regulators in TB monocytes, *IRF1* and *STAT1* (*Figure 5a*, top panel), were the only genes in common between the TB monocyte gene signature (*Berry et al., 2010*), the 'IL6/STAT3 pathway'

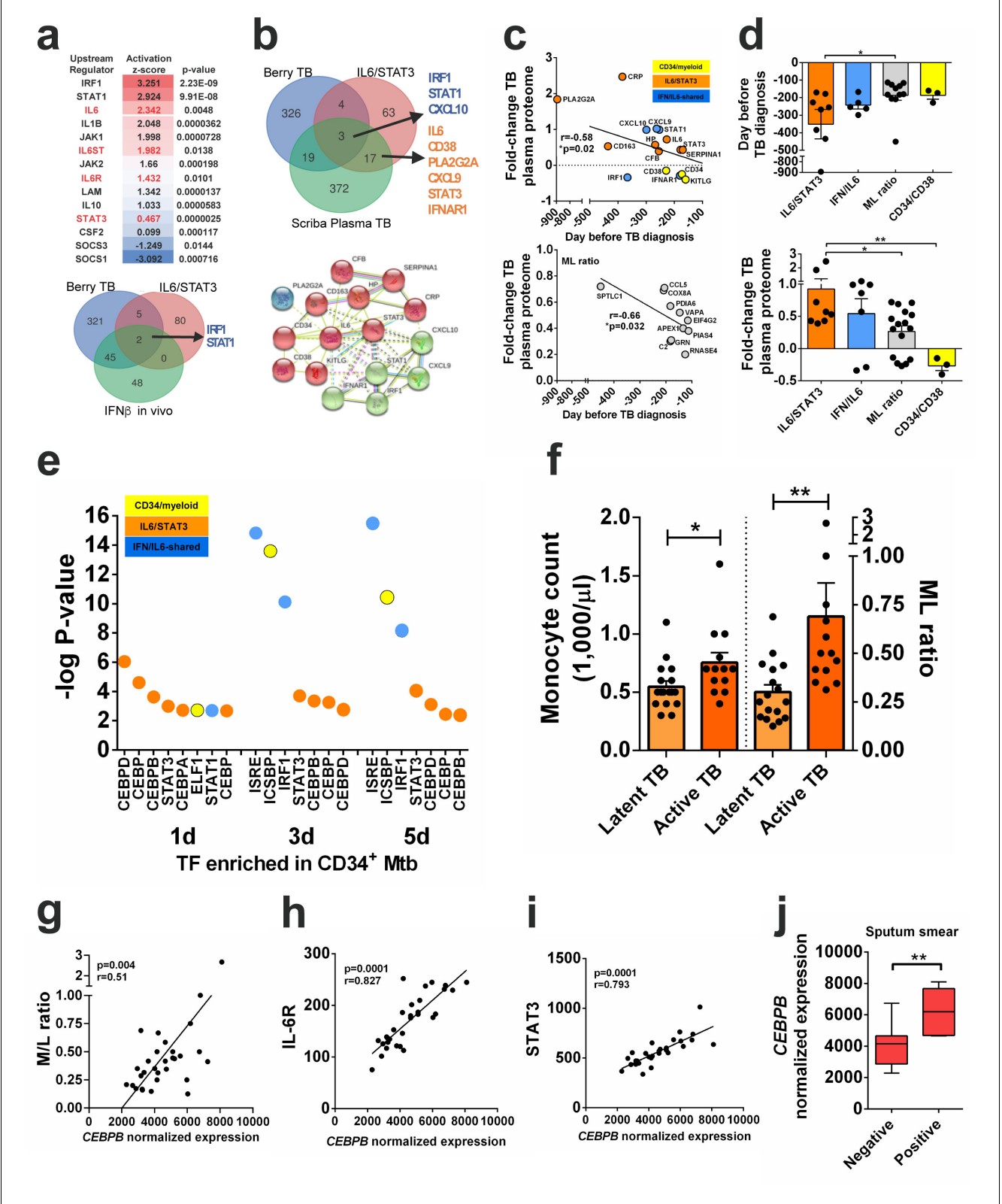

**Figure 5.** *IL6/IL6R/CEBPB* gene module is enriched in active TB transcriptome and proteome and correlates with monocyte expansion. (a) Top panel: upstream regulators significantly enriched by causal Ingenuity Pathway Analysis (IPA) in monocyte transcriptomes from patients with active TB (GSE19443), ranked by activation z-score, p-values are corrected for genome-wide testing (FDR). Bottom panel: IRF1 and STAT1 are the top upstream regulators shared between the 'Berry TB' disease signature (*Berry et al., 2010*) (GSE19435, GSE19439, GSE19444), the 'IL6/STAT3' pathway (Hallmark

Figure 5 continued

GSEA) and the human 'in vivo IFN-β" signature (GSEA HECKER_IFNB1_TARGETS). (b) Top panel: overlap between the 'Berry TB' disease signature, the 'IL6/STAT3' pathway and the 'Scriba plasma TB' proteomic signature (*Scriba et al., 2017*) identified 'IFN/IL6-shared' and 'IL6/STAT3-specific' signatures. Bottom panel: significant STRING protein-protein interaction network (p<10$^{-16}$) for 'IFN/IL6-shared' genes (green marbles) and 'IL6/STAT3' genes (red marbles), clustering separately by k-means. (c) Top panel: significant linear increase over time before active TB diagnosis in plasma proteome (*Scriba et al., 2017*) for 'CD34/myeloid' (yellow), 'IL6/STAT3' (orange) and 'IFN/IL6-shared' (blue) clusters found in (b). Bottom panel: monocyte/lymphocyte (ML) ratio gene set members defined by *Naranbhai et al. (2015)* over time before active TB diagnosis in plasma proteome (*Scriba et al., 2017*).(d) Top panel: increased 'IL6/STAT3' cluster protein expression precedes monocyte expansion markers (ML ratio gene set) in the TB plasma proteome. Bottom panel: data as in d) shows significant higher fold-changes for 'IL6/STAT3' vs. 'ML ratio' or 'CD34/myeloid' cluster members. *p-value<0.05, ** p-value<0.01. (e) Transcription factor enrichment analysis (GSEA) of differentially expressed genes determined by RNA-seq in *Mtb*-exposed CD34$^+$ cells in vitro (n = 3 donors). (f), monocyte count and ML ratio in samples from latent vs active TB patients from *Berry et al. (2010)* reanalysis. *p-value<0.05, ** p-value<0.01 between active TB vs latent TB groups. Transcriptional data of whole blood reanalysis from *Berry et al. (2010)* shows a significant correlation of *CEBPB* transcripts with g) M/L ratio; h) *IL6R*; i) *STAT3* transcript levels, and j) mycobacterial positivity in sputum smears in patients with active TB. ** p-value<0.01 between positive vs negative groups.
DOI: https://doi.org/10.7554/eLife.47013.019

The following source data and figure supplement are available for figure 5:

**Source data 1.** Raw data from *Figure 5*.
DOI: https://doi.org/10.7554/eLife.47013.021

**Figure supplement 1.** Gene expression and protein conservation of the IFN/IL6/CEBP gene module and correlation analysis to TB disease.
DOI: https://doi.org/10.7554/eLife.47013.020

and the 'in vivo IFN-β" signature (*Figure 5a*, Venn diagram, bottom panel), suggesting these genes might be regulated by both IL-6 and type I IFN during active TB in vivo. Since type I IFN and IL-6 share the ability to induce phosphorylation of both STAT1 and STAT3 (*Ho and Ivashkiv, 2006*), we ran gene set enrichment analysis (GSEA) (*Subramanian et al., 2005*) to identify potential overlapping downstream target genes in the whole blood 'Berry TB' disease signature (*Berry et al., 2010*). In addition to the previously demonstrated type I IFN/STAT1 signature (*Berry et al., 2010*), the 'IL6/STAT3' pathway was significantly enriched in this data set (FDR-corrected p<10$^{-4}$, *Supplementary file 2*). Next, we defined protein signatures by overlapping the 'Berry TB', the 'IL6/STAT3' pathway with a published plasma proteome defining disease progression from latent to active TB (*Scriba et al., 2017*) ('Scriba Plasma TB', *Figure 5b*, Venn Diagram top panel). STRING network analysis of protein-protein interactions confirmed two clusters (*Figure 5b*, bottom panel), which comprised three signatures: 'CD34/myeloid', 'IL6/STAT3' and 'IFN/IL6-shared' pathways. Reanalysis of the published 'Scriba Plasma TB' proteome set (*Scriba et al., 2017*) confirmed increased IL-6/STAT3 protein levels and changes in CD34/CD38 homeostasis, which were found to be early events in TB pathogenesis (*Figure 5c*, top panel). 'IL6/STAT3' pathway-associated proteins such as PLA2G2A, CRP, STAT3, IL-6 and CFB increased around 12 months before TB diagnosis (*Figure 5c,d*, top panels - orange circles and bars), which was concomitant with significant changes in the 'IFN/IL6-shared' plasma markers CXCL10, IFNAR1 and MMP9 (*Figure 5c,d* top panels - blue circles and bars). The 'IL6/STAT3' and 'IFN/IL6-shared"pathways were also significantly enriched in a gene set recently linked to monocyte expansion in vivo, measured as monocyte:lymphocyte (ML) ratio (*Naranbhai et al., 2015*) (*Supplementary file 2*), and positively correlated with mycobacterial growth in vitro, thus connecting monocyte expansion and increased *Mtb* survival. Interestingly, the changes found in both the 'IL6/STAT3' and 'IFN/IL6-shared' pathways during development of TB disease preceded enrichment of the 'ML ratio' gene set (*Naranbhai et al., 2015*) (6 months before diagnosis, p<0.05, *Figure 5c*, bottom panel and *Figure 5d*, top panel, gray bar) and reduction of the CD34/CD38 gene markers, in agreement with our in vitro model of *Mtb*-enhanced CD34$^+$ differentiation (CD34$^+$ → CD34$^+$CD38$^+$ → CD14$^+$). Moreover, fold changes were higher for 'IL6/STAT3' pathway genes than for 'IFN/IL6-shared' genes, and significantly higher than 'ML ratio' genes (p<0.05) or 'CD34/myeloid' differentiation genes (p<0.01) (*Figure 5d* bottom panel). Together, these data suggest sequential activation of IL-6/IL-6R and IFN signaling pathways before monocyte expansion during TB disease progression in vivo, raising a possible link between these two events in disease pathogenesis. In support of this idea, CD34$^+$ cells exposed to *Mtb* in vitro displayed increased levels of pSTAT1 as well as C/EBPβ (and a slight enhancement of C/EBPα), which are key TF regulators of ISGs and myeloid differentiation genes, respectively (*Figure 5—figure supplement*

*1a*). Interestingly, qPCR experiments from HK *Mtb*-exposed CD34$^+$ cells did not show induction of myeloid differentiation TFs *CEBPA* and *CEBPB* (*Figure 4—figure supplement 1g*). Furthermore, we observed that *CEBPB, CEBPD and STAT3* as well as *IRF1, STAT1 and ICSBP/IRF8* TFs were significantly enriched in *Mtb*-infected CD34$^+$ transcriptomes (*Figure 5e* and *Figure 5—figure supplement 1b* - *Figure 2—source data 1*), which were associated with increased mycobacterial replication in vitro (*Figure 1e*). These results suggest that *Mtb* infection activates a gene module shared by both type I IFN and IL-6, linking downstream ISGs and CEBPs.

## An *IL6/IL6R/CEBP* gene module correlates with monocyte expansion and TB severity

TB pathogenesis is a convoluted process which interconnects mycobacterial dissemination, host inflammatory responses and systemic tissue pathology. To further investigate a potential link between this gene module (*IL6/IL6R/CEBP*) with disease severity and monocyte expansion in vivo, we first examined large transcriptomic data sets of 'disseminated TB', which includes extrapulmonary and lymph node TB (GSE63548). The 'IL6/STAT3' pathway was found to be significantly enriched among differentially expressed genes in both extrapulmonary (FDR p=10$^{-3}$) and lymph node TB (FDR p=10$^{-4}$, *Supplementary file 2*). Furthermore, downstream targets of *STAT3, CEBPB, CEBPD, SPI1/PU1, ICSBP/IRF8,* which are TF regulators of myeloid differentiation, were enriched in 'disseminated TB' and in the 'ML ratio' gene sets (*Supplementary file 2*), suggesting these TFs are activated during severe disease and associated to monocyte expansion in vivo. In contrast, ISRE (STAT1/STAT2) and IRF1 motifs, the major upstream regulators observed in TB monocyte transcriptome (*Figure 5a* top panel) and shared between IL-6 and IFN signaling (*Figure 5b*), were not enriched in the 'ML ratio' gene set (*Supplementary file 2*). Of note, only *CEBPB* targets were significantly enriched in the 'ML ratio' gene set in healthy subjects (*Supplementary file 2*), supporting its link with myeloid differentiation during homeostasis. Since the *IL6/IL6R/CEBP* gene module was correlated with both systemic disease dissemination and monocyte expansion, two processes associated with TB disease (*Rogers, 1928*; *Schmitt et al., 1977*), we next examined whether these genes might be connected to disease severity in a published cohort with detailed clinical parameters and transcriptome data (*Berry et al., 2010*). When compared to latent TB subjects, we observed that both the monocyte counts and ML ratio were significantly increased in active TB patients (*Figure 5f*). *CEBPB* transcripts positively correlated with ML ratio levels (*Figure 5g*), *IL6R* transcripts (*Figure 5h*), *STAT3* (*Figure 5i*) as well as inflammatory biomarkers such as C-reactive protein (CRP) and erythrocyte sedimentation rate (ESR) (*Figure 5—figure supplement 1c*). In addition, *CEBPB* mRNA levels were significantly higher in *Mtb*-positive vs. *Mtb*-negative sputum smears (*Figure 5j*), and positively correlated to tissue damage, total symptom counts as well as *ISG15* levels (*Figure 5—figure supplement 1c*), in agreement with our previous findings (*Dos Santos et al., 2018*). Taken together, these results indicate that the *IL6/IL6R/CEBP* gene module is a hub correlated with monocyte expansion during *Mtb* infection in vivo and is amplified in severe pulmonary and systemic disease.

## Recent mammalian/primate genetic changes link an IFN/IL-6/IL-6R/CEBP axis to monocyte expansion and TB pathogenesis in humans

The 'type I IFN' signature found in active TB (*Berry et al., 2010*), shared with the IL-6/IL-6R-regulated gene set (*Figure 5*), comprises a number of well-characterized ISGs with cross-species antiviral activity such as *IRF1* and *OAS*. It has been reported that these ISGs have been undergoing strong purifying selection during primate evolution (*Manry et al., 2011*; *Shaw et al., 2017*), including recent Neanderthal introgression (*Enard and Petrov, 2018*; *Quach et al., 2016*). We thus undertook an evolutionary approach to investigate whether the *IL6/IL6R/CEBP* gene module and its partial overlap with type I IFN signaling is linked to monocyte expansion and TB severity. To do so, we performed a stepwise analysis, starting from early mammalian emergence (>100 million years ago (mya), over primate (>50 mya) and hominid evolution (>15 mya). We have also examined the recent human evolution including Neanderthal introgression (<100,000 years ago) and human pathogen adaptation (15,000–1,500 years ago), up to extant human genetic variation through analysis of large genome wide association studies (GWAS).

First, STRING network measurements of amino acid conservation and gene co-occurrence across mammalian and primate evolution revealed that IL-6, IL-6R and C/EBP family members C/EBPα, C/

EBPβ and C/EBPδ differ substantially throughout primate evolution and even among closely related hominins (*Pan troglodytes* and *Gorilla gorilla*) (*Figure 6a*, heat map). In contrast, matched control molecules in the same STRING network (KLF5/NFKB1/MAPK1//STAT1/STAT3) remained largely conserved in most mammals, and even in birds and reptiles (*Figure 6a*, heat map and *Figure 5—figure supplement 1d*). Next, to investigate the biological consequence of the evolutionary differences in overlapping IL-6/IFN signaling, we reanalyzed cross-species type I IFN regulation from the 'mammalian interferome' database (*Shaw et al., 2017*). As expected, the conserved 'IFN/IL6-shared' genes *CXCL10/CXCL9/STAT1/STAT2* displayed higher fold-changes upon type I IFN treatment across all 10 species (from chicken to human, *Figure 6b*, top panel). Interestingly, the 'IL-6/STAT3' pathway genes *IL6, STAT3* and *SOCS3* were also significantly upregulated while *IL6R* was significantly downregulated (*Figure 6b*, top panel) in the same experimental setting. Among CD34/myeloid differentiation genes, *ICSBP/IRF8* and *CD38* were strongly upregulated, but only in 4/10 and 2/10 species, respectively, while *ELF1* was homogeneously and significantly upregulated in 9/10 species (*Figure 6b*, top panel). These results suggest that type I IFN consistently regulates expression of IL-6 signaling and myeloid-associated genes in different species. However, among the entire *IL6/IL6R/CEBP* myeloid gene set, *CEBPB* was the topmost variable ISG across mammalian evolution (CV >1000%, *Figure 6b*, bottom panel). Remarkably, type I IFN-induced upregulation of both *CEBPB* and *CEBPD*, previously identified as NF-IL6 and NF-IL6β, respectively (*Ramji and Foka, 2002*) was present only in humans and lacking in all other mammals investigated (*Figure 6b*, bottom panel, inset). Mechanistically, ChipSeq analysis of IFN-treated human CD14$^+$ monocytes corresponding to regions with active chromatin (DNase Hypersensitivity Sites, DHS and H3K27 acetylation, not shown) confirmed the existence of functional STAT1 peaks in *CEBPB* and *CEBPD* (*Figure 6c*, top and middle panels, denoted by vertical blue lines). These peaks correlated with increased downstream transcription in CD14$^+$ cells, as compared to purified CD34$^+$ cells (*Figure 6c*, RNA-seq). In agreement with our findings (represented in *Figure 6b*, bottom panel, *inset*), only 3 out of 11 (27%) STAT1 binding peaks in *CEBPB* and *CEBPD* were found in conserved regions (Conservation Birds-Mammals line, *Figure 6c*), while 6 out of 7 (86%) STAT1 peaks were conserved in *CXCL9* and *CXCL10* genes (*Figure 6c*, bottom panel). Interestingly, transcriptional regulation of *CEBPB* and *CEBPD* in humans and macaques, but not mouse cells stimulated with double-stranded RNA, which mimics a viral infection (*Figure 6—figure supplement 1a*), were also observed in an independent data set (*Hagai et al., 2018*). As expected, *CXCL9* and *CXCL10* responses are conserved in dsRNA-stimulated cells from humans, macaques and mouse (*Figure 6—figure supplement 1a*). Thus, transcriptional induction of *CEBPB* and *CEBPD* controlled by IL-6- and type I IFN-signaling appears as a relatively recent event in mammalian and primate evolution.

## Genome-wide association studies (GWAS) connect the *IL6/IL6R/CEBP* gene module with monocyte expansion in TB disease

Since genetic susceptibility and transcriptional responses to intracellular pathogens have shown significant links to Neanderthal introgression in populations of European and Asian descent (*Dannemann et al., 2017*; *Quach et al., 2016*), we next explored enrichment for introgression in the *IL6/IL6R/CEBPB* and 'ML ratio' gene sets. As shown in *Figure 6d* (*Figure 2—source data 1*), eleven genes with Neanderthal introgression were significantly upregulated in our *Mtb*-exposed CD34$^+$ cells transcriptome (enrichment p<0.0001). Of those, *OAS1, OAS2* and *MT2A* transcripts had significantly higher effect sizes upon ML ratios, as compared to other introgressed genes (p<0.05) and to all other genes shown to regulate ML ratio in vivo (p<0.001, *Figure 6d*). This finding was confirmed in a recently published data set (n = 198) of purified microbial-exposed CD14$^+$ monocytes from a Belgian cohort of European (EUB) and African (AFB) descendance, with documented presence or absence of Neanderthal introgression, respectively (*Quach et al., 2016*). Strikingly, 9 out of 11 introgressed genes enriched during *Mtb*-triggered monocyte differentiation (*Figure 6d*, Venn diagram) were significantly upregulated in TLR1/TLR2-stimulated (*Figure 6d*, right panel), but not unstimulated monocytes (not shown). These findings suggest pathogen exposure may enhance gene pathways recently selected during hominid evolution linked to monocyte expansion.

We next interrogated whether the *IL6/IL6R/CEBP* gene module was linked with monocyte expansion in several large published data sets of standing human variation. Two large GWAS studies (*Astle et al., 2016*; *Kanai et al., 2018*) containing >230,000 individuals have identified single-nucleotide polymorphisms (SNPs) in or adjacent to *IL6R, CEBPA-CEBPD-CEBPE* and *ICSBP/IRF8* genes as

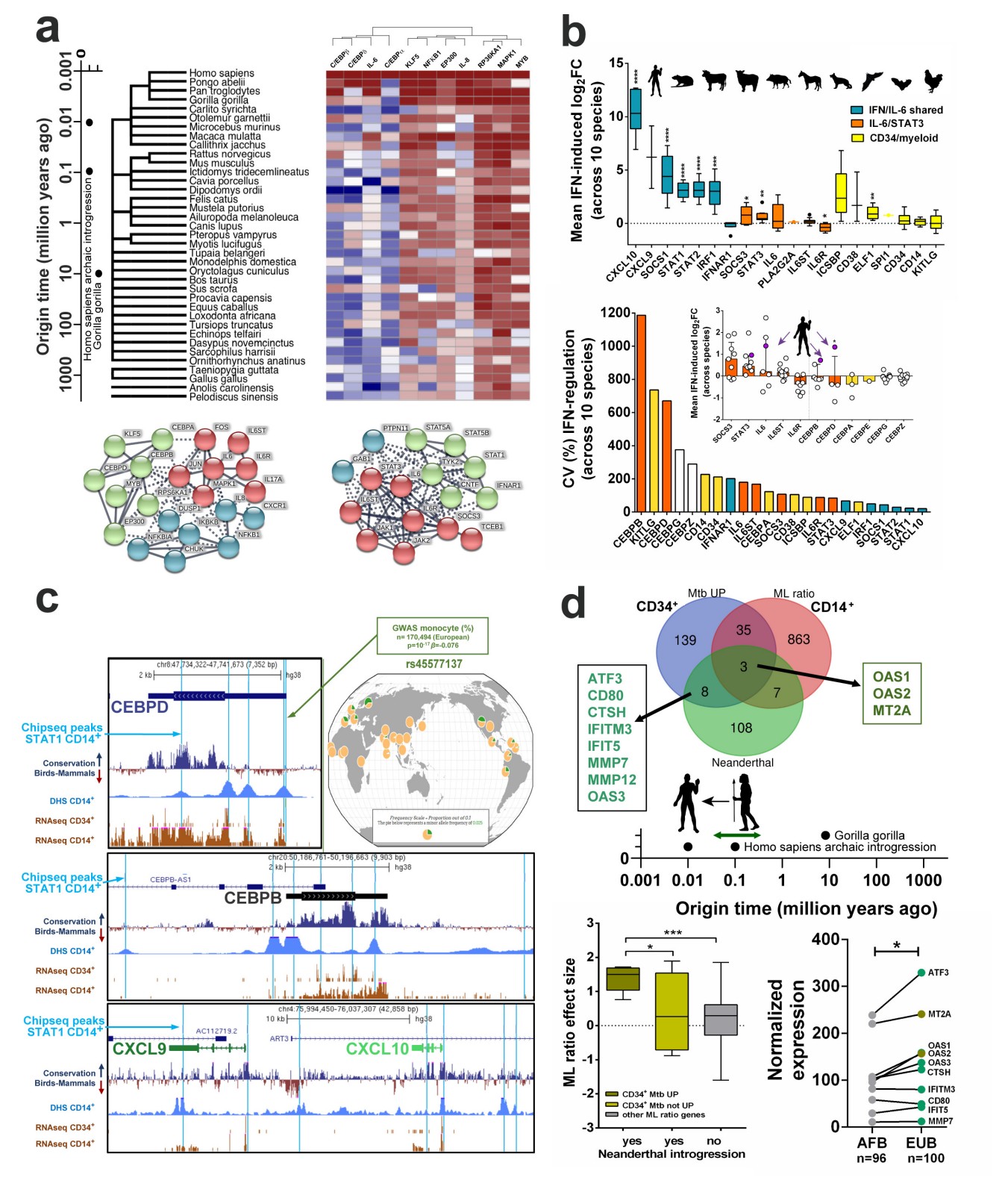

**Figure 6.** Evolutionary recent and human-specific genetic adaptation link *IL6/IL6R/CEBP* gene module with monocyte expansion and TB pathogenesis. (a) Heat map showing *CEBPB* network generated by STRING co-occurrence protein conservation scores across primates, mammals, birds and reptiles. Note only *CEBPB* and *CEBPA* differ strongly among hominids, while *CEBPA/CEBPB/CEBPD* vary significantly throughout primate and mammalian evolution, as compared to highly conserved *STAT1/STAT3* (*Figure 5—figure supplement 1d*). (b) Top panel: Highly conserved type I IFN upregulation

*Figure 6 continued*

of 'IFN/IL6-shared' genes from humans to birds (derived from http://isg.data.cvr.ac.uk/) (*Shaw et al., 2017*), as compared to 'IL6/STAT3' and 'CD34/ myeloid differentiation' genes. Bottom panel: *CEBPB* and *CEBPD* displays highest variation, and *CXCL10* the lowest variation in type I IFN transcriptional regulation across human-mammalian-bird evolution. *Inset, CEBPB* and *CEBPD* selectively acquired type I IFN upregulation in humans (filled circles); ** p-value<0.01 and * p-value<0.05 represent *CEBPB* and *CEBPD* values, respectively for humans versus the other species. (c) ChipSeq analysis of STAT1-binding peaks in *CEBPD* (top panel), *CEBPB* (middle panel), *CXCL9* and *CXCL10* (bottom panel) in IFN-stimulated human monocytes, corresponding to regions with active chromatin (DNase Hypersensitivity Sites, DHS) and correlating with increased downstream transcription in CD14[+] monocytes, as compared to purified CD34[+] cells. Conservation analysis among >40 vertebrates (phyloP [*Pollard et al., 2010*], from chicken to human, analogous to *Figure 5b*) indicates STAT1 peaks are mostly conserved in *CXCL9/CXCL10* (6/7) but not in *CEBPD/CEBPB* (3/11). (d) Top panel: overlap between human genes with significant Neanderthal introgression (*Enard and Petrov, 2018*; *Quach et al., 2016*), genes differentially expressed in *Mtb*-exposed CD34[+] cells (CD34[+] Mtb UP) and the 'ML ratio' gene set. Bottom left panel: *OAS1, OAS2* and *MT2A* transcripts presented significantly higher effect sizes upon ML ratios, corresponding to monocyte expansion, as compared to other introgressed genes (p<0.05) and to all other genes shown to regulate ML ratio in vivo (p<0.001). Bottom right panel: normalized expression of introgressed genes found in CD34[+]Mtb UP (Venn diagram) in TLR1/2 agonist-treated monocytes from a cohort of matched Belgian individuals of European (EUB) vs. African (AFB) descendance, with documented presence or absence of Neanderthal introgression (*Quach et al., 2016*), respectively. p-value<0.05, ** p-value<0.01, *** p-value<0.001, **** p-value<0.0001.
DOI: https://doi.org/10.7554/eLife.47013.022

The following source data and figure supplements are available for figure 6:

**Source data 1.** Raw data from *Figure 6*.
DOI: https://doi.org/10.7554/eLife.47013.025
**Figure supplement 1.** TB susceptibility genes of the IFN/IL6/CEBP gene module and ISG induction during myeloid differentiation in vitro.
DOI: https://doi.org/10.7554/eLife.47013.023
**Figure supplement 1—source data 1.** Raw data from *Figure 6—figure supplement 1*.
DOI: https://doi.org/10.7554/eLife.47013.024

significantly associated to blood monocyte counts (*Supplementary file 2* and ranked in *Figure 7a* as monocyte count GWAS). Moreover, gene-specific z-scores for human polygenic adaptation to pathogens in 51 different populations worldwide (*Daub et al., 2013*) were positive, representing higher levels of population differentiation, for all genes in our proposed *IL6/IL6R/CEBP* myeloid differentiation module (except *CXCL10*, *Figure 7a* and *Supplementary file 3*). Lastly, we examined whether myeloid differentiation genes identified in this study are found in GWAS TB susceptibility genes. A significant enrichment (p<0.0001) for differentially expressed genes from our *Mtb*-exposed CD34[+] transcriptome and TB susceptibility GWAS/candidate genes (18 out of 172 genes, including *IL6, STAT1* and *CD14)* was also observed (*Figure 6—figure supplement 1b*). Similarly, a significant (p<0.0001) overlap was found for the *IL6/IL6R/CEPB* module and TB genetic susceptibility (six shared genes *CD14, CXCL10, IL6, IL6R, IRF1* and *STAT1*, *Figure 6—figure supplement 1b*).

As ranked in *Figure 7a*, 24 out of 28 members of this gene module display a genome-wide, transcriptomic, proteomic or functional association to human TB, being strongest for *IL6* and its downstream signaling TFs *CEBPB* and *CEBPD*, demonstrated in 5–6 independent data sets each. Collectively, our multi-level-based evidence suggests *Mtb* exploits an evolutionary recent IFN/IL-6/IL-6R/CEBP axis linked to monocyte expansion and human TB disease.

## Discussion

Emerging evidence has suggested that *Mtb* establishes an infectious niche in the human bone marrow during active TB, which is associated with altered numbers of leukocytes in the periphery (*Das et al., 2013*; *Mert et al., 2001*; *Naranbhai et al., 2015*; *Rogers, 1928*; *Schmitt et al., 1977*; *Tornack et al., 2017*; *Wang et al., 2015*). In the present study, we observed that *Mtb* consistently stimulated myeloid differentiation molecules in CD34[+] cell cultures from three different human tissues, namely: bone marrow, peripheral blood or cord blood samples. Employing a purified cord-blood-derived CD34[+] culture cell system, we observed that *Mtb* enhances IL-6R-mediated myeloid differentiation by human primary CD34[+] cells in vitro. Importantly, IL-6/IL-6R downstream molecules such as C/EBPβ, C/EBPδ, STAT3 and their targets were significantly enriched in cell transcriptomes from active TB patients as well as were positively correlated with disease severity. Therefore, our data expands previous studies and raise a scenario in which *Mtb* skews myeloid development, mediated by IL-6/IL-6R signaling, as a key step in human TB pathogenesis.

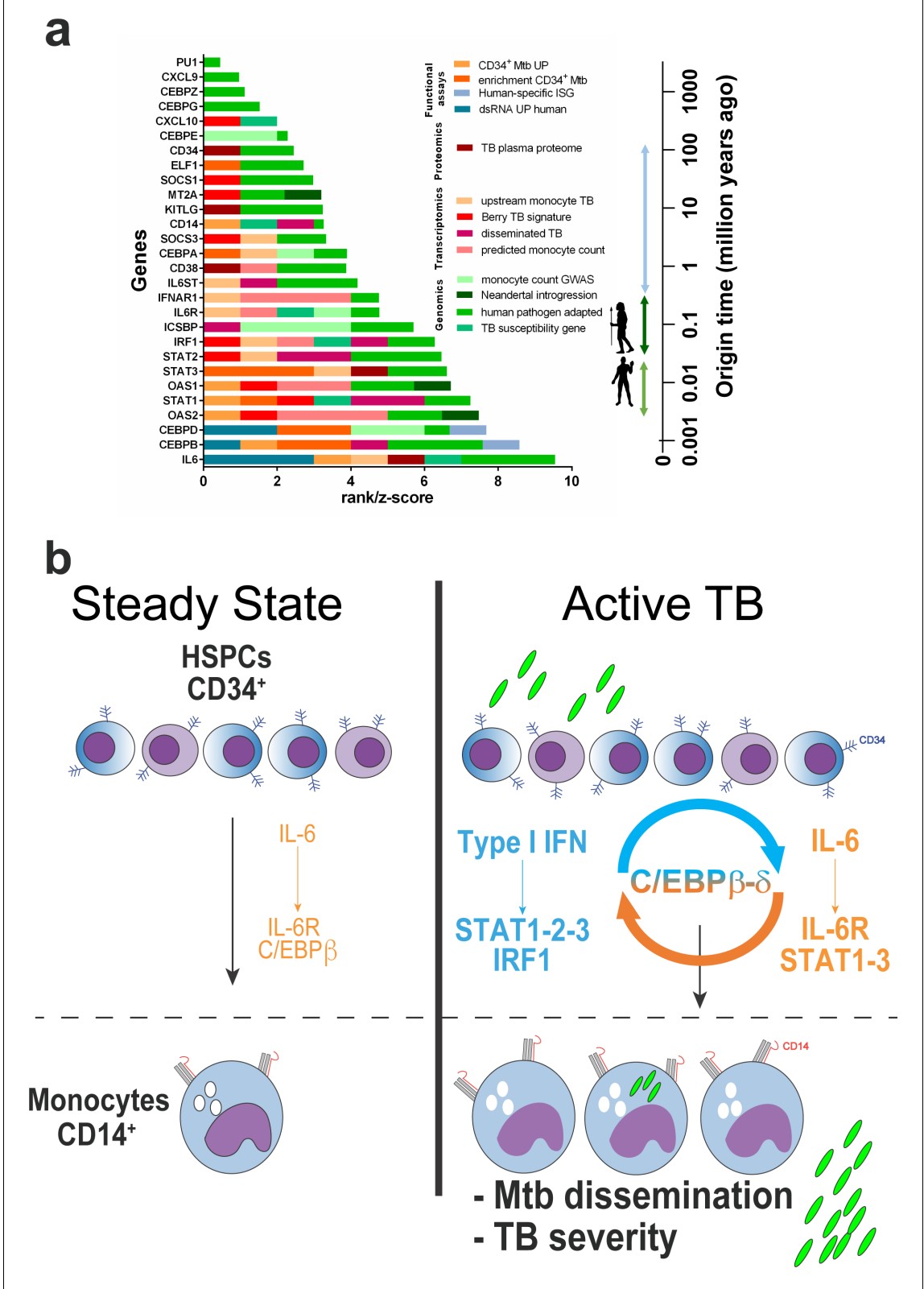

**Figure 7.** Compiled multi-level evidence for an *IL6/IL6R/CEBP* gene module linking CD34[+] myeloid differentiation to TB pathogenesis and disease severity. (a) Ranks and scores were determined as 0–1 (presence-absence in data set) or 0-1-2-3, according to enrichment analysis or differential gene expression (quartiles); z-scores were obtained from Daub et al.[55] (b) Proposed model for C/EBPβ and C/EBPδ acting as a bridge in the type I IFN and IL-6 feed-forward loop exploited by *Mtb* to induce monocyte differentiation and TB disease severity (details in the text).

*Figure 7 continued on next page*

*Figure 7 continued*

DOI: https://doi.org/10.7554/eLife.47013.026

The following source data is available for figure 7:

**Source data 1.** Raw data from *Figure 7*.

DOI: https://doi.org/10.7554/eLife.47013.027

While *Mtb* enhanced *IL6* expression in purified CD34$^+$ cell cultures from all donors, *IFNA* and *IFNB* mRNA were detected in some but not all donors. However, ISGs were highly enriched in the bacteria-exposed samples suggesting that although low/undetectable amounts of type I IFN were produced in infected cell cultures (*Rodero et al., 2017*), these cytokines were present in the cell culture (*Figure 4—figure supplement 1h*). Furthermore, our results show that live *Mtb* is a potent stimulus to induce ISGs (*Figure 6—figure supplement 1c - Figure 2—source data 1*) and myeloid differentiation in primary human CD34$^+$ cells. However, while heat killed mycobacteria induced IL-6 production by CD34$^+$ cells, it poorly stimulated *STAT1*, *CEBPB* and differentiation cell surface molecules by progenitor cells as well as CD14$^+$ monocyte levels. Interestingly, although IL-6R signaling was involved in both myeloid differentiation and *Mtb* growth by CD34$^+$ cell cultures, type I or type II IFN signaling were not. These results suggest that monocyte maturation is connected to *Mtb* proliferation in vitro and could explain why the effects of anti-IL6R antibodies on cellular differentiation inhibited bacteria growth (*Figure 4f–i*). Although the ISG gene set was enriched in *Mtb*-exposed CD34$^+$ cells, our data suggest type I or type II IFN signaling appear not to mediate *Mtb*-enhanced monocyte development in vitro. Collectively, this evidence suggests that unknown activities of live pathogen infection regulate myeloid differentiation involving an IL-6R-mediated process and implies cross-talking of regulatory pathways between live *Mtb*, IL-6 and IFN signaling to boost myeloid differentiation of CD34$^+$ cells. At the molecular level, it has been reported that IFN-induced C/EBPβ triggers gamma-activated transcriptional elements (GATE) sequences independent of STAT1 (*Li et al., 2007*), suggesting the existence of cooperative and/or redundant roles of IL-6 and IFN signaling in different molecular settings. The mechanisms by which endogenous IL-6, IFN-α/β and live *Mtb* interplay to enhance C/EBP-mediated myeloid differentiation of HSPCs require further investigation. Of note, in our previously characterized cohort of multiple sclerosis patients (*Menezes et al., 2014*; *Van Weyenbergh et al., 2001*), IFN-β therapy in vivo did not significantly change monocyte or lymphocyte counts, nor did it increase the ML ratio after three months of treatment in patients with documented clinical response (data not shown), supporting the idea that type I IFN by itself is not sufficient to cause monocyte expansion in vivo.

It has not been determined how mature myeloid cell populations from active TB patients acquire the 'ISG' signature. While monocytes and other cells may encounter *Mtb*-associated inflammatory stimuli in infected tissues (e.g. lungs and liver), our data suggest the possibility that *Mtb* may activate these cells during their development in the bone marrow. We have not directly addressed whether circulating monocytes/granulocytes acquire their phenotype in the bone marrow, during development of myeloid progenitors in vivo. Nevertheless, a recent study by *Norris and Ernst (2018)* demonstrated increased monocyte egress from the bone marrow in a murine model of *Mtb* infection. Considering mycobacteria (*Arts et al., 2018*; *Das et al., 2013*; *Joosten et al., 2018*; *Mert et al., 2001*; *Mitroulis et al., 2018*) can access the bone marrow and stimulate IL-6, it is possible that individuals draining higher amounts of *Mtb* into the bone marrow display increased inflammatory alterations including IL-6R-mediated myelopoiesis, upregulation of IFN-stimulated responses and amplified disease severity. Likewise, we found that enrichment of the *IL6/IL6R/CEBP* axis positively correlated with systemic disease such as lymph node and extrapulmonary TB (*Figure 5f,g*).

Several independent studies have also indicated a detrimental role of 'IFN and IFN-induced genes' during *Mtb* infection in human TB (*Berry et al., 2010*; *Bustamante et al., 2014*; *Dos Santos et al., 2018*; *Novikov et al., 2011*; *Scriba et al., 2017*; *Zhang et al., 2018*) and murine models (*Antonelli et al., 2010*; *Manca et al., 2001*). Our results expand these previous studies, revealing a novel *IL6/IL6R/CEBP* gene module and its link to monocyte development, mycobacterial dissemination and TB disease severity. Furthermore, as evidenced by *Scriba et al. (2017)* and re-analyzed in the present study (*Figure 5c,d*), both IFN and IL6 pathways are early events in TB pathogenesis, detectable in the plasma proteome >6 months before diagnosis. Nevertheless, a pivotal role for IL6/

IL6R signaling has not been evident from previous 'omics' approaches. While IL-6 signaling partially overlaps with type I IFN responses (*Figure 5a,b*), possibly due to their shared ability to activate STAT1 and STAT3 (*Ho and Ivashkiv, 2006*), whole blood transcriptomic analyses in TB are predominated by an 'IFN-inducible neutrophil signature' (*Berry et al., 2010*). Therefore, the high numbers of neutrophils in the blood possibly mask differential expression of other gene sets in less frequent populations, such as monocytes, monocyte subsets and, in particular, Lin⁻CD34⁺ cells.

Most ISGs present in the whole blood Berry TB signature (*Berry et al., 2010*), the Scriba TB plasma proteome (*Scriba et al., 2017*) and the Naranbhai et al. ML ratio gene set (*Naranbhai et al., 2015*) displayed cross-species type I IFN-induction throughout mammalian evolution (*Figure 6b*). Despite a shared STAT1/STAT3 activation by type I IFN and IL-6, homeostatic activation of C/EBPβ is mostly IL-6-specific, as evidenced by data mining and STRING analysis (*Figure 6a*), in keeping with its original description as NF-IL6 (*Akira et al., 1990*). Across species, *CEBPB* and *CEBPD* are among the highly variable ISG of the entire IFN/IL-6/CD34/myeloid gene set (*Figure 6b*). Likewise, although type I IFN-induced *IL6* and *STAT3* transcription are conserved in all mammalian species studied, IFN-inducibility of *CEBPB* and *CEBPD* mRNA appeared to be recently acquired in primate evolution. In line with our observations, *IL6*, *CEBPB* and *SPI1/PU1* genetic polymorphisms have been previously associated with TB susceptibility (*Zhang et al., 2014*; *Zhang et al., 2012*). In addition, a recently identified *trans* e-QTL (rs5743618) (*Quach et al., 2016*) in *TLR1*, a gene with peak Neanderthal introgression (*Dannemann et al., 2017*; *Enard and Petrov, 2018*; *Hagai et al., 2018*; *Quach et al., 2016*) has been associated to TB susceptibility in several populations worldwide (*Barletta-Naveca et al., 2018*; *Naderi et al., 2016*; *Qi et al., 2015*). Biological pathway analysis of genes significantly regulated in trans of rs5743618 revealed a significant enrichment of IL-6/STAT3 signaling and *IL6* as the most connected gene (data not shown). In agreement, *Mtb*-induced macrophage IL-6 production, among other cytokines, can be predicted on the basis of strong genetic components as recently reported by *Bakker et al. (2018)* in a large GWAS/immunophenotyping cohort study. By employing a data-driven multi-level analysis from large cohorts, we expand these previous observations and revealed significant genetic links shared between IL-6/IL-6R/CEBP signaling, CD34⁺ myeloid differentiation, monocyte homeostasis and TB susceptibility (compiled in *Figure 7a*). Together, these findings favor the hypothesis that such genetic changes have undergone stepwise mammalian, primate and recent human selection, including Neanderthal introgression and worldwide population-specific pathogen adaptation.

In summary, our observations suggest that *Mtb* boosts myeloid differentiation by exploiting a feed-forward loop between IL-6 and type I IFN molecular networks, bridged by C/EBPβ (and C/EBPδ) (*Figure 7b*). Yet, further experiments will define the precise mechanisms of crosstalk between IFN, IL-6 and CEBP family members, specifically CEBPβ and CEBPδ, during natural *Mtb* infection. While this question merits direct investigation, nonetheless, the use of IL-6R blockade as an adjunct therapy to treat multi-drug resistant severe TB has been proposed (*Okada et al., 2011*; *Zumla et al., 2016*). Thus, the present study provides evidence of a novel host-directed target for therapeutic intervention in a major human disease.

# Materials and methods

## Key resources table

| Reagent type (species) or resources | Designation | Source or reference | Identifiers | Additional information |
|---|---|---|---|---|
| Strain (*Mycobacterium tuberculosis*) | H37Rv | ATCC | | |
| Strain (*Mycobacterium tuberculosis*) | Mtb-CS267 | Clinical Isolate | This study | |

*Continued on next page*

Continued

| Reagent type (species) or resources | Designation | Source or reference | Identifiers | Additional information |
|---|---|---|---|---|
| Cell Line (*Homo sapiens*) | Human Cord Blood (CB) purified CD34$^+$ cells | STEMCELL Technologies | Catalog#70008.5 | Cell line maintained in StemSpan II Expansion Media - STEMCELL Technologies-Catalog#09605 |
| Cell (*Homo sapiens*) | Peripheral blood mononuclear cell (PBMC) | | | Cells maintained in RPMI 1640 complete, Sigma-Aldrich – Catalog#R8758 |
| Chemical Compound, drug | Middlebrook 7H10 agar | BD Biosciences | Catalog# 262710 | Supplemented with 10% Oleic Acid,Albumin, Dextrose, Catalase (Sigma-Aldrich – Catalog# M0678-1VL) |
| Chemical Compound, drug | L-Glutamine (200 mM) | Sigma-Aldrich | Catalog# 25030081 | |
| Chemical Compound, drug | Sodium Pyruvate (100 mM) | Life Technologies. | Catalog# 11360070 | |
| Biological Sample | Genomic DNA, *Mycobacterium tuberculosis*, Strain H37Rv | This study | | Dosage: 10 µg/ml |
| Chemical Compound, drug | *Mycobacterium tuberculosis*, Strain H37Rv, Purified Lipoarabinomannan (LAM) | BEI Resources | Catalog# NR-14848 | Dosage: 5 µg/mL |
| Peptide, Recombinant protein | Recombinant Human Interleukin 6 (rh IL-) | ImmunoTools | Catalog# 11340064 | |
| Antibody | FITC Anti-Lineage 1, human: CD3 clone SK7, CD16 clone 3G8, CD19 clone SJ25C1, CD20 clone L27, CD14 clone MφP9, CD56 clone NCAM. | BD Biosciences | Catalog# 340546 | (1:30) |
| Antibody | PE Anti-human CD34, clone 581. | BD Biosciences | Catalog# 555822 | (1:20) |
| Antibody | FITC Anti-human CD34, clone 8G12. | BD Biosciences | Catalog# 345801 | (1:50) |
| Antibody | PerCP Anti-human CD34, clone 581. | Biolegend | Catalog# 343519 | (1:20) |
| Antibody | PECy7 Anti-human HLA-DR, clone L243 | Biolegend | Catalog# 307615 | (1:200) |
| Antibody | Bv510 Mouse Anti-Human HLA-DR, clone G46-6 | BD Horizon | Catalog# 563083 | (1:100) |
| Antibody | APC Anti-human CD38, clone HIT2 | Biolegend | Catalog# 303510 | (1:100) |
| Antibody | Bv421 Anti-Human CD64, clone 10.1. | BD Biosciences | Catalog# 562872 | (1:100) |
| Antibody | FITC Anti-human CD10, clone HI10A | BD Biosciences | Catalog# 340925 | (1:50) |
| Antibody | Alexa Fluor 488 Anti-human CD14, clone M5E2. | Biolegend | Catalog# 301817 | (1:50) |

*Continued on next page*

*Continued*

| Reagent type (species) or resources | Designation | Source or reference | Identifiers | Additional information |
|---|---|---|---|---|
| Antibody | APCCy7 Anti-mouse/human CD11b, clone M1/70 | Biolegend | Catalog# 101226 | (1:100) |
| Antibody | V450 Anti-human CD14, clone MφP9 | BD Biosciences | Catalog# 560350 | (1:100) |
| Antibody | PE Anti-human CD66b, clone G10F5. | Biolegend | Catalog#305106 | (1:100) |
| Antibody | APCCy7 Anti-human BDCA1, clone L161. | Biolegend | Catalog# 331520 | (1:200) |
| Antibody | FITC Anti-human CD41a, clone 6C9. | ImmunoTools | Catalog# 21330413 | (1:50) |
| Antibody | APC Anti-human BDCA2, clone 201A | Biolegend | Catalog# 354205 | (1:50) |
| Antibody | Bv510 Anti-human BDCA3, clone 1A4. | BD Biosciences | Catalog# 563298 | (1:100) |
| Antibody | PE Anti-human CD123, clone 7G3. | BD Biosciences | Catalog# 554529 | (1:20) |
| Antibody | APC Anti-human CD16, clone 3G8. | BD BiosciencesCatalog# | 561248 | (1:50) |
| Antibody | V450 Anti-human CD64, clone 10.1. | BD Biosciences | Catalog# 561202 | (1:20) |
| Antibody | FITC Anti-human CD3, clone UCHT1. | Biolegend | Catalog# 300440 | (1:100) |
| Antibody | FITC Anti-human CD19, clone 4G7. | BD Biosciences | Catalog# 347543 | (1:50) |
| Antibody | Alexa Fluor 488 Anti-human CD14, clone M5E2. | BD Biosciences | Catalog# 561706 | (1:50) |
| Antibody | PerCP-Cy5.5 Anti-human CD34, clone 8G12. | BD Biosciences | Catalog# 347203 | (1:25) |
| Antibody | PE Anti-human IL-6, clone 8C9. | ImmunoTools | Catalog# 21670064 | (1:10) |
| Antibody | FITC Anti-human CD56, clone NCAM16.2. | BD Biosciences | Catalog# 345811 | (1:100) |
| Antibody | FITC Anti-human CD16, clone HI16a. | ImmunoTools | Catalog# 21810163 | (1:100) |
| Antibody | Monoclonal Anti-STAT1 (phospo Y701), clone M135. | Abcam | Catalog# ab29045 | (1:1000) |
| Antibody | Monoclonal Anti-STAT1, clone SM1. | Abcam | Catalog# ab3987 | (1:1000) |
| Antibody | Polyclonal Anti-C/EBPβ | Santa Cruz Biotechnology | Catalog# sc-150 | (1:250) |
| Antibody | Neutralizing Anti-human IFNAR2, clone MMHAR-2 | PBL Assay Science | Catalog# 21370–1 | Dosage: 1 µg/mL |

*Continued on next page*

*Continued*

| Reagent type (species) or resources | Designation | Source or reference | Identifiers | Additional information |
|---|---|---|---|---|
| Antibody | Monoclonal, Anti-IFN-γ, clone B27. | ImmunoTools | Catalog# 21853531 | Dosage: 10 µg/mL |
| Antibody | Anti-IL6, Tocilizumab. | Roche | | Dosage: 1 µg/mL |
| Antibody | Monoclonal Anti-beta actin | Abcam | Catalog# mAbcam 8226 | (1:5000) |
| Chemical Compound, drug | Flexible Viability Stain 450 | BD Horizon | Catalog#562247 | (1:1000) |
| Chemical Compound, drug | Carbol Fuchsin | Sigma-Aldrich | Catalog# C4165 | |
| Chemical Compound, drug | Methylene Blue | Sigma-Aldrich | Catalog# 03978 | |
| Chemical Compound, drug | Hoechst 33342 | Immunochemistry technologies | Catalog# 639 | |
| Commercial assay, kit | M-PER Mammalian Protein Extraction Reagent | Thermo Fisher Scientific | Catalog# 78501 | |
| Commercial assay, kit | cOmplete ULTRA Tablets, Mini, EASYpack Protease Inhibitor Cocktail | Roche | Catalog# 05 892970001 | |
| Commercial assay, kit | High-Capacity cDNA Reverse Transcription Kit | Applied Biosystems | Catalog# 4368814 | |
| Chemical Compound, drug | TRIzol LS Reagent | Invitrogen | Catalog# 10296010 | |
| Commercial assay, kit | NuGEN - Trio low input RNA-seq | NuGEN | Catalog#0507–08 | |
| Software, algorithm | FlowJo software v. 10.1 | TreeStar | FlowJo, RRID:SCR_008520_ | https://www. flowjo.com/ |
| Software, algorithm | GraphPad Prism 6 Software | GraphPad | GraphPad Prism, RRID:SCR_002798 | https://www. graphpad.com/ |

## Reagents

*Mtb* Ara-LAM was obtained from BEI Resources and used at 5 µg/mL. *Mtb* H37Rv genomic DNA was obtained from 28 days colonies growing in Löwenstein–Jensen medium by CTAB method as previously described (*Yamashiro et al., 2016*). Recombinant human (*rh*) IL-6 was purchased from Immunotools. Anti-IFNAR2A (clone MMHAR-2, PBL) and anti-IFN-γ (clone B27, Immunotools) neutralizing antibodies were used at 1 and 10 µg/mL, respectively and anti-IL-6R (Tocilizumab, Roche) was used at 1 µg/mL. Fluorescent dye Syto24 was obtained from Thermo Fisher Scientific.

## Mycobacteria cultures

The virulent laboratory H37Rv *Mtb* strain and the clinical *Mtb* isolate (Mtb-CS267) were maintained in safety containment facilities at LACEN and UFSC as described elsewhere (*Yamashiro et al., 2016*). Briefly, *Mtb* was cultured in Löwenstein-Jensen medium (Laborclin) and incubated for 4 weeks at 37°C. Prior to use, bacterial suspensions were prepared by disruption in saline solution using sterile glass beads. Bacterial concentration was determined by a number 1 McFarland scale, corresponding to $3 \times 10^8$ bacteria/mL.

## Subjects samples, cells and *Mtb* infections

This study was approved by the institutional review boards of Universidade Federal de Santa Catarina and The University Hospital Prof. Polydoro Ernani de São Thiago (IRB# 89894417.8.0000.0121). Informed consent was obtained from all subjects. Peripheral blood and bone marrow mononuclear

cells were obtained using Ficoll-Paque (GE) in accordance with the manufacturer's instructions. Briefly, blood collected in lithium-heparin containing tubes was further diluted in saline solution 1:1 and added over one volume of Ficoll-Paque reagent. The gradient was centrifuged for 40 min at 400 x $g$, 20°C. The top serum fraction was carefully removed, the mononuclear fraction was harvested and washed once in a final volume of 50 mL of saline solution for 10 min at 400 x $g$, 20°C. Subsequently, cell pellet was suspended and washed twice with 20 mL of saline solution for 10 min at 200 x $g$, 20°C, to remove platelets. Cells were then suspended to the desired concentration in RPMI 1640 (Life Technologies) supplemented with 1% fresh complement inactivated (30 min at 56°C) autologous serum, 2 mM $L$-glutamine (Life Technologies), 1 mM sodium pyruvate (Life Technologies) and 25 mM HEPES (Life Technologies). Human Cord Blood (CB) purified CD34$^+$ cells from five different donors were obtained from STEMCELL Technologies and resuspended in StemSpan Expansion Media – SFEM II (STEMCELL Technologies) according to manufacturer's instruction. Optimal cell density for replication was 5 × 10$^4$ CD34$^+$ cell/mL. In a set of experiments, CD34+ cells were further enriched using a cell sorter (FACSMelody, BD). Following 4 days of expansion, cells were washed and diluted in SFEM II media without cytokine cocktail to the desired concentration. Culture purity was assessed by FACS and showed more than 90% of CD34$^+$ events after expansion. For in vitro infection experiments, 1 McFarland scale was diluted in media to fit the desired multiplicity of infection (MOI). For each experiment, bacteria solution was plated in Middlebrook 7H10 agar (BD Biosciences) supplemented with 10% Oleic Acid Albumin Dextrose Complex (OADC) and incubated at 37°C to confirm initial bacteria input. In a set of experiments, 1 McFarland scale was incubated with 500 nM of Syto24 dye as described previously (*Yamashiro et al., 2016*). In some experiments, H37Rv *Mtb* was heat killed (HK) at 100°C for 30 min. *Leishmania infantum* promastigotes were kindly provide by Ms. Karime Mansur/UFSC and Dr. Patrícia Stoco/UFSC and used at MOI = 3. In cytokine/cytokine neutralizing experiments, cells were pretreated with anti-IFNAR2 (1 µg/mL), anti-IFN-γ (10 µg/mL) or anti-IL-6R (1 µg/mL) for 1 hr and exposed to *Mtb* (MOI3). Following different time points post-infection, cells were harvested and centrifuged at 400 x $g$ for 10 min, 20°C. Supernatants were then stored at −20°C, cells washed once in sterile saline solution and lysed by using 200 µL of 0.05% Tween 80 solution (Vetec) in sterile saline. Cell lysates were diluted in several concentrations (10$^{-1}$ to 10$^{-5}$), plated onto Middlebrook 7H10 agar (BD Biosciences) supplemented with OADC 10% and incubated at 37°C. After 28 days, colony-forming units (CFU) were counted and the results were expressed graphically as CFU/mL.

## Microscopy experiments

After different time points post-infection, cells were washed and fixed with PFA 2% overnight at 4°C. Subsequently, cells were washed with sterile water solution and adhered into coverslips by cytospin centrifugation. Samples were then fixed with methanol for 5 min, washed with sterile water and stained with carbol-fuchsin (Sigma) for 2 min. Samples were washed once with sterile water and counterstaining was done with methylene blue dye (Sigma) for 30 s. Coverslips were fixed in slides with Permount mounting medium (Sigma) and examined using Olympus BX40 microscope and digital camera Olympus DP72. Quantification was performed by enumeration of number of infected cells or "cytoplasm-rich cells, defined as cells bigger than 10 um and with approximately 2:1 cytoplasm/nucleus ratio. Cells were counted in at least 10 fields from two different experiments and plotted as % of events. Syto24-stained *Mtb* was visualized in CD34$^+$ cells by using confocal fluorescence-equipped inverted phase contrast microscope and photographed with a digital imaging system camera. Briefly, 1 × 10$^5$ CD34$^+$ cells were seeded in 24-well plate and infected with *Mtb* syto24, MOI3, for 4 hr. Further, cells were washed, fixed with PFA 2% and adhered into coverslip by cytospin centrifugation. For nucleus visualization, cells were stained with Hoechst 33342 (Immunochemistry technologies) for 2 hr. Cells were after washed and mounted for analysis in Leica DMI6000 B confocal microscope.

## Immunoblotting

CD34$^+$ cells were seeded at 3 × 10$^5$ cells in 24-well plate and infected with *Mtb* (MOI3). After 5 days of infection, cells were centrifuged at 4°C, pellet was lysed using M-PER lysis buffer (Thermo Fisher Scientific) containing protease inhibitors (Complete, Mini Protease Inhibitor Tablets, Roche) and protein extracts were prepared according to manufacturer's instructions. For Western blot, 15

µg of total protein were separated and transferred to nitrocellulose difluoride 0.22 µm blotting membranes. Membranes were blocked for 1 hr with TBST containing 5% w/v BSA and subsequently washed three times with TBST for 5 min each wash. Further, membranes were then probed with anti-pSTAT1 Y701 1:1000 (M135 – Abcam), anti-STAT1 1:1000 (SM1 – Abcam), anti-C/EBPβ 1:250 (sc-150 – Santa Cruz) or anti-β-actin 1:5000 (8226 – Abcam) primary antibodies diluted in 5% w/v BSA, 0.1% tween 20 in TBS, at 4°C with gentle shaking overnight. Membranes were washed with TBST, incubated in secondary HRP-linked Ab for 2 hr at room temperature, washed and chemiluminescence developed using ECL substrate (Pierce). Relative expression was normalized with β-actin control and pixel area was calculated using ImageJ software.

## Flow cytometry

PBMC and bone marrow mononuclear cells were seeded at $5 \times 10^5$ cells per well in a final volume of 200 µL. After 4 hr of resting at 37°C with 5% $CO_2$, cells were infected with Mtb (MOI3) for 72 hr, unless indicated otherwise. Cells were detached from the plate by vigorous pipetting, centrifuged at 450 x g for 10 min and washed twice in saline solution and stained with fixable viability stain FVS V450 (BD Biosciences) at the concentration 1:1000 for 15 min at room temperature. Cells were then washed with FACS buffer (PBS supplemented with 1% BSA and 0.1% sodium azide) and incubated with 10% pooled AB human serum at 4°C for 15 min. The following antibodies were used in different combinations for staining:

Staining of human CD34$^+$ in PBMC: anti-Lin1(CD3, CD14, CD16, CD19, CD20,CD56) (FITC, clones MφP9, NCAM 16, 3G8, SK7, L27, SJ25-C1), anti-CD34 (PE, PE, clone 581), anti-CD34 (FITC, 8G12), anti-CD34 (PerCP, clone 581), anti-HLA-DR (PE-Cy7, clone L243), anti-HLA-DR (Bv510, clone G46-6), anti-CD38 (APC, clone HIT2), anti-CD4 (APC-Cy7,GK1.5), anti-CD64 (Bv421, clone 10.1), anti-CD10 (FITC, clone HI10A), anti-CD14 (V450, clone MoP9), anti-CD14 (Alexa488, clone M5E2) were added at titrated determined concentration and incubated for 40 min at 4°C.

Staining of CB CD34$^+$ cells: anti-CD34 (PE, clone 581), anti-CD11b (APCCy7, clone M1/70), anti-CD4 (APC-Cy7, clone GK1.5), anti-CD64, (Bv421, MoP9), anti-CD14 (V450, clone MoP9) anti-CD14 (Alexa488, clone M5E2), anti-CD66b (PE, clone G10F5), anti-BDCA1 (APC-Cy7, clone L161), anti-CD41a (FITC, clone 6C9), anti-BDCA2 (APC, clone 201A), anti-BDCA3 (Bv510, clone 1A4), anti-Clec9A (A700, clone FAB6049P), anti-CD123 (PE, clone 7G3), anti-CD16 (APC, clone 3G8) were added at titrated determined concentrations and incubated for 40 min at 4°C. In a set of experiments, PBMCs were exposed to live Mtb, HK Mtb or LPS (100 ng/mL) for 24 hr and the Golgi Plug protein transport inhibitor (BD Biosciences) was added for the last 6 hr according to manufacturer's instructions. Then, cells were surface stained with FITC-Lin (FITC-anti-CD3, Alexa Fluor 488-anti-CD14, FITC-anti-CD16, FITC-anti-CD19, FITC-anti-CD56) and PerCP/Cy5.5-anti-CD34, followed by permeabilization and PE-anti-IL-6 (clone 8C9) staining. All cells were subsequently washed with FACS buffer and resuspended in 2% PFA. Cells were acquired on BD FACS Verse with FACSuite software. Analysis were performed using FlowJo software v. 10.1 (TreeStar).

## Real-time quantitative PCR

Total RNA was extracted from CD34$^+$ cells exposed or not with Mtb. RNA was extracted after 1, 3 and 5 days of infection using TRIzol reagent (Thermo) according to manufacturer's instruction. Using 1 µg of RNA, cDNA was produced with a High-Capacity cDNA Reverse Transcription Kit (Applied Biosystems) and 2 µL of 1:8 diluted product was used to the quantitative PCR reaction in a final volume of 10 µL. qPCR reactions were performed using the primers for: IFNA2A F: 5'-TTGACCTTTGC TTTACTGGT-3', R: 5'-CACAAGGGCTGTATTTCT TC-3'. IL6 F: 5'- CCACACAGACAGCCACTCAC-3', R: 5'-AGGTTGTTTTCTGCCAGTGC-3'. IFNB F: 5'- AAACTCATGAGCAGTCTGCA-3', R: 5'-AGGAGA TCTTCAGTTTCGGAGG-3'. IFNG F: 5'- TCAGCTCTGCATCGTTTTGG-3', R: 5'-GTTTCCATTA TCCGCTACATCTGAA-3'. IFI16 F: 5'-ACTGAGTACAACAAAGCCATTTGA-3', R: 5'-TTGTGACATTG TCCTGTCCCCAC-3'. MX1 F: 5'-ATCCTGGGATTTTGGGGCTT-3', R: 5'-CCGCTTGTCGCTGGTG TCG-3'. ISG15 F: 5'-TCCTGGTGAGGAATAACAAGGG-3', R: 5'-CTCAGCCAGAACAGGTCGTC-3'. CXCL8 F: 5'-GAGGTGATTGAGGTGGACCAC-3', R: 5'-CACACCTCTGCACCCAGTTT-3'. IL1RA F: 5'-ATGGAGGGAAGATGTGCCTGTC-3', R: 5'-GTCCTGCTTTCTGTTCTCGCTC-3'. GRB2 F: 5'-GAAA TGCTTAGCAAACAGCGGCA-3', R: 5'-TCCACTTCGGAGCACCTTGAAG-3'. STAT1 F: 5'-ATGGCAG TCTGGCGGCTGAATT-3', R: 5'-CCAAACCAGGCTGGCACAATTG-3'. CEBPA F: 5'-

TGGACAAGAACAGCAACGAGTA-3', R: 5'-ATTGTCACTGGTCAGCTCCAG-3'. *CEBPB* F: 5'-TGGGACCCAGCATGTCTC-3', R: 5'-TCCGCCTCGTAGTAGAAGTTG-3'.

## RNA isolation and sequencing

Total RNA from purified CB CD34$^+$ cells exposed to *Mtb* in vitro was isolated using TRIzol LS (Invitrogen; 10296010). RNA-seq libraries were prepared using the Nugen Ovation Trio low input RNA Library Systems V2 (Nugen; 0507–08) according to the manufacturer's instructions by the Nucleomics Platform (VIB, Leuven, Belgium). Pooled libraries were sequenced as 150 bp, paired-end reads on an Illumina HiSeq 2500 using v4 chemistry.

## RNA-seq data quality assessment and differential expression analyses

Illumina sequencing adapters and reads with Phred quality scores lower than 20 were removed with Trimmomatic (0.36). Trimmed reads were aligned to *H. sapiens* reference genome (hg38) by STAR (2.6.0 c). Aligned reads were mapped to genes using feature Counts from the Subread package (1.6.1). Genes with reads of less than three were removed. Library based normalization was used to transform raw counts to RPKM and further normalized using the edgeR TMM normalization (3.10.0). Data were then transformed using the limma voom function (3.36.2), prior to batch correction using ComBat (sva 3.28.0). Negative binomial and linear model-based methods were used for differential expression analysis, using packages edgeR and limma packages. Differentially expressed genes (DEGs) were calculated with t-statistics, moderated F-statistic, and log-odds of differential expression by empirical Bayes moderation of the standard errors (*Supplementary file 4*).

## CellNet and CellRouter analysis

We applied CellNet to classify RNA-seq samples as previously described (*Cahan et al., 2014*). Raw RNA sequencing data files were used for CellNet analysis. We used R version 3.4.1, CellNet version 0.0.0.9000, Salmon (*Patro et al., 2017*) version 0.8.2 and the corresponding index downloaded from the CellNet website. We used CellRouter (*Lummertz da Rocha et al., 2018*) to calculate signature scores for each sample based on cell-type specific transcriptional factors collected from literature. Specifically, for this analysis, we normalized raw counts by library size as implemented in the R package DESeq2 (*Love et al., 2014*). We then plotted the distributions of signature scores across experimental conditions. Moreover, we used CellRouter to identify genes preferentially expressed in each experimental condition and used those genes for Reactome pathways enrichment analysis using the Enrichr package version 1.0.

## Systems biology analysis

Ingenuity Pathway Analysis (IPA) software was used to perform the initial pathway/function level analysis on genes determined to be differentially expressed in transcriptomic analysis (Ingenuity Systems, Red Wood City, CA). Uncorrected p-values and absolute fold-changes were used with cut-offs of p<0.05 (monocyte transcriptomes from active TB patients) or p<0.01 (differentially expressed genes in *Mtb*-exposed CD34$^+$ cells and all publicly available datasets from GEO). Differentially expressed genes were sorted into gene networks and canonical pathways, and significantly overrepresented pathways and upstream regulators were identified. Additional pathway, GO (Gene Ontology) and transcription factor target enrichment analysis was performed using GSEA (Gene Set Enrichment Analysis, Broad Institute Molecular Signatures Database (MSigDB)) and WebGestalt (WEB-based GEne SeT AnaLysis Toolkit). Gene sets from GO, Hallmark, KEGG pathways, WikiPathways and Pathway Commons databases, as well as transcription factor motifs, were considered overrepresented if their FDR-corrected p-value was <0.05. To validate our compiled *IL6/IL6R/CEBP* and CD34$^+$ myeloid differentiation gene modules, we used STRING (version 10.5) protein-protein interaction enrichment analysis (www.string-db.org), using the whole human genome as background. Principal component analysis, correlation matrices, unsupervised hierarchical (Eucledian distance) clustering were performed using XLSTAT and visualized using MORPHEUS (https://software.broadinstitute.org/morpheus/). Chipseq, active chromatin and transcriptional (RNAseq) data of CD14 and CD34+ cells were downloaded from ENCODE (https://genome.ucsc.edu/ENCODE/) and visualized using the UCSC browser (*Haeussler et al., 2019*).

## Data processing and statistical analyses

Data derived from in vitro experiments was processed using GraphPad Prism six software and analyzed using unpaired $t$ test, one-way ANOVA or two-way ANOVA according to the experimental settings. Data from experiments performed in triplicate are expressed as mean ± SEM. Non-parametric tests (Mann-Whitney, Spearman correlation) were used for clinical data (sputum bacillar load, modal X-ray grade, symptom count) and molecular data that were not normally distributed, Pearson correlation was used for molecular data with a normal distribution. A list of the statistics analysis methods used in each figure is available a supplementary file (*Supplementary file 4*). Statistical significance was expressed as follows: *$p \leq 0.05$, **$p \leq 0.01$ and ***$p \leq 0.001$.

## Acknowledgements

We thank Drs. José Henrique M Oliveira/UFSC and João T Marques/UFMG for their critical reading of this manuscript and UFSC microscopy (LCME) and biology (LAMEB) facilities for technical support. This work was funded by Howard Hughes Medical Institute – Early Career Scientist (AB; 55007412), National Institutes of Health Global Research Initiative Program (AB, TW008276), Coordenação de Aperfeiçoamento de Pessoal de Nível Superior (CAPES) Computational Biology (DSM; 23038.010048/2013–27), FWO (JVW; G0D6817N), FWO (TD; VLAIO IWT141614) and CNPQ/PQ Scholars (AB and DSM).

## Additional information

### Funding

| Funder | Grant reference number | Author |
|---|---|---|
| Howard Hughes Medical Institute | Early Career Scientist 55007412 | André Báfica |
| Coordenação de Aperfeiçoamento de Pessoal de Nível Superior | 23038.010048/2013-27 | Daniel S Mansur |
| Fonds Wetenschappelijk Onderzoek | G0D6817N | Johan Van Weyenbergh |
| National Institutes of Health | Global Research Initiative Program TW008276 | André Báfica |
| Conselho Nacional de Desenvolvimento Científico e Tecnológico | PQ | André Báfica Daniel S Mansur |
| Fonds Wetenschappelijk Onderzoek | VLAIO IWT141614 | Tim Dierckx |

The funders had no role in study design, data collection and interpretation, or the decision to submit the work for publication.

### Author contributions

Murilo Delgobo, Conceptualization, Data curation, Formal analysis, Investigation, Methodology, Writing—original draft; Daniel AGB Mendes, Conceptualization, Data curation, Formal analysis, Methodology, Writing—review and editing; Edgar Kozlova, Tim Dierckx, Data curation, Software, Formal analysis, Methodology; Edroaldo Lummertz Rocha, Gabriela F Rodrigues-Luiz, Conceptualization, Data curation, Software, Formal analysis, Investigation, Methodology; Lucas Mascarin, Greicy Dias, Daniel O Patrício, Data curation, Formal analysis, Methodology; Maíra A Bicca, Gaëlle Bretton, Yonne Karoline Tenório de Menezes, Márick R Starick, Formal analysis, Methodology; Darcita Rovaris, Joanita Del Moral, Resources, Methodology; Daniel S Mansur, Formal analysis, Funding acquisition, Visualization, Methodology, Writing—review and editing; Johan Van Weyenbergh, Conceptualization, Data curation, Formal analysis, Supervision, Funding acquisition, Investigation, Methodology, Writing—original draft, Writing—review and editing; André Báfica, Conceptualization, Data curation, Formal analysis, Supervision, Funding acquisition, Investigation,

Visualization, Methodology, Writing—original draft, Project administration, Writing—review and editing

### Author ORCIDs
André Báfica https://orcid.org/0000-0002-5148-600X

### Ethics
Human subjects: This study was approved by the institutional review boards of Universidade Federal de Santa Catarina and The University Hospital Prof. Polydoro Ernani de São Thiago (IRB# 89894417.8.0000.0121). Informed consent was obtained from all subjects.

### Decision letter and Author response
Decision letter https://doi.org/10.7554/eLife.47013.044
Author response https://doi.org/10.7554/eLife.47013.045

## Additional files

### Supplementary files
• Supplementary file 1. Reactome Pathways analysis of *Mtb*-exposed and control CD34+ cell transcriptomes.
DOI: https://doi.org/10.7554/eLife.47013.028

• Supplementary file 2. Systems analysis (Ingenuity Pathway Analysis and Gene Set Enrichment Analysis) of cohorts of healthy controls, patients with latent TB, active TB, disseminated TB, overlap with IL6/STAT3 signaling and myeloid development.
DOI: https://doi.org/10.7554/eLife.47013.029

• Supplementary file 3. Human adaptation z-scores for IL6/IL6R/CEBP CD34 myeloid gene module and Gene set enrichment of Top500 human adaptation genes.
DOI: https://doi.org/10.7554/eLife.47013.030

• Supplementary file 4. List of statistical methods used in the manuscript.
DOI: https://doi.org/10.7554/eLife.47013.031

• Transparent reporting form DOI: https://doi.org/10.7554/eLife.47013.032

### Data availability
Sequencing data have been deposited in GEO under accession code GSE129270.

The following previously published datasets were used:

| Author(s) | Year | Dataset title | Dataset URL | Database and Identifier |
|---|---|---|---|---|
| Maji A, Misra R, Mondal AK, Singh Y | 2015 | Expression profiling of lymph nodes in tuberculosis patients reveal inflammatory milieu at site of infection | https://www.ncbi.nlm. nih.gov/geo/query/acc. cgi?acc=GSE63548 | NCBI Gene Expression Omnibus, GSE63548 |
| Berry MP, Graham CM, McNab FW, Xu Z, Bloch SA, Oni T, Wilkinson KA, Banchereau R, Skinner J, Wilkinson RJ, Quinn C, Blankenship D, Dhawan R, Cush JJ, Mejias A, Ramilo O, Kon OM, Pascual V, Banchereau J, Chaussabel D, O'Garra A | 2010 | Blood Transcriptional Profiles of Active TB (UK Test Set Separated) | https://www.ncbi.nlm. nih.gov/geo/query/acc. cgi?acc=GSE19443 | NCBI Gene Expression Omnibus, GSE19443 |
| Berry MP, Graham CM, McNab FW, | 2010 | Transcriptional profiles in Blood of patients with Tuberculosis - | https://www.ncbi.nlm. nih.gov/geo/query/acc. | NCBI Gene Expression Omnibus, |

| Xu Z, Bloch SA, Oni T, Wilkinson KA, Banchereau R, Skinner J, Wilkinson RJ, Quinn C, Blankenship D, Dhawan R, Cush JJ, Mejias A, Ramilo O, Kon OM, Pascual V, Banchereau J, Chaussabel D, O'Garra A | | Longitudinal Study | cgi?acc=GSE19435 | GSE19435 |
|---|---|---|---|---|
| Berry MP, Graham CM, McNab FW, Xu Z, Bloch SA, Oni T, Wilkinson KA, Banchereau R, Skinner J, Wilkinson RJ, Quinn C, Blankenship D, Dhawan R, Cush JJ, Mejias A, Ramilo O, Kon OM, Pascual V, Banchereau J, Chaussabel D, O'Garra A | 2010 | Blood Transcriptional Profiles in Active and Latent Tuberculosis UK (Training Set) | https://www.ncbi.nlm.nih.gov/geo/query/acc.cgi?acc=GSE19439 | NCBI Gene Expression Omnibus, GSE19439 |
| Berry MP, Graham CM, McNab FW, Xu Z, Bloch SA, Oni T, Wilkinson KA, Banchereau R, Skinner J, Wilkinson RJ, Quinn C, Blankenship D, Dhawan R, Cush JJ, Mejias A, Ramilo O, Kon OM, Pascual V, Banchereau J, Chaussabel D, O'Garra A | 2010 | Blood Transcriptional Profiles of Active and Latent TB (UK Test Set) | https://www.ncbi.nlm.nih.gov/geo/query/acc.cgi?acc=GSE19444 | NCBI Gene Expression Omnibus, GSE19444 |

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
