## [Decision Letter]

[Editors’ note: this article was originally rejected after discussions between the reviewers, but the authors were invited to resubmit after an appeal against the decision.]

Thank you for submitting your work entitled "*M. tuberculosis* hijacks an evolutionary recent IFN-IL-6-CEBP axis linked to monocyte expansion and disease severity" for consideration by *eLife*. Your article has been reviewed by four peer reviewers, one of whom is a member of our Board of Reviewing Editors, and the evaluation has been overseen by a Senior Editor. The following individuals involved in review of your submission have agreed to reveal their identity: Ruslan Medhzitov (Reviewer #2).

Our decision has been reached after consultation between the reviewers. Based on these discussions and the individual reviews below, we regret to inform you that your work will not be considered further for publication in *eLife*.

Reviewers recognized that your manuscript provides interesting evidence illustrating that *Mycobacterium tuberculosis* is able to drive monocyte differentiation from human hematopoietic stem cells through the IL-6/CEBP axis. The evolutionary description of this process is also interesting. However, there are numerous problems with some of the data, both conceptual and technical, which were identified by the reviewers. Further, the establishment of causality is somewhat lacking.

Reviewer #1:

Background: *M. tuberculosis* has been detected in the bone marrow – suggestive of a possible niche for tubercle bacteria in active and latent TB. This observation also points towards a role for mycobacteria in driving differentiation of certain cell types from Hematopoietic stem cells (HSCs).

Key findings include:

1) The demonstration of the ability of tubercle bacteria to replicate in Hematopoietic stem cell progenitors (HSCPs), in this case CD14^+^ and CD34^+^ cells.

2) Transcriptional analysis of infected CD13^+^ cells indicated that *M. tuberculosis* infection drives these cells towards myeloid differentiation, further confirmed by flow cytometry. This observation was confirmed with cells from cord blood also.

3) A role for IL6R was confirmed in this myeloid differentiation pathway, using a neutralizing antibody.

4) By interrogating transcriptome data, the authors confirm that the IL6R-CEBP pathway is central in monocyte expansion.

5) Comparative genomics and evolutionary analysis reveal that this pathway has been the subject of recent selection.

Major Concerns:

1) Figure 1—figure supplement 1C is meant to indicate that low numbers of bacteria associate with CD34^+^ cells. However, there is no comparison with CD14^+^, no quantification no statistics. This figure does not have use as it stands. Similarly, with Figure 1—figure supplement 1D – one assumes the control here would be uninfected cells? In this case, the statistical comparison shown is meaningless – comparing the bacterial load in CD14^+^ cells versus CD34^+^ cells on one graph would be more useful. The authors indicate they are comparable – but need to provide the data in a way that allows for this comparison.

2) Figures 1 F and G need quantification, followed by statistical comparison.

3) Figure 2—figure supplement 1B and C needs statistics, – are the differences significant, – if not the text needs to be revised to indicate this. This applies to all supplementary graphs, – these should have statistics. If differences are not significant "NS" should be stated. Same with Figures in the main text.

4) The comparative genomics/transcriptomics and evolutionary analysis is very dense, – it needs to be simplified and presented in a clearer manner.

Reviewer #2:

In this manuscript Delgobo et al. demonstrate a novel role for *Mtb* in driving monocyte differentiation from human hematopoietic stem cells via an IL-6/CEBP axis. In vitro studies show that *Mtb* replicates in human CD34^+^ cells, and causes upregulation of the monocyte marker CD14 as well as enrichment of genes associated with myeloid differentiation. This in vitro monocyte expansion is demonstrated to be IL-6 dependent and associated with STAT1 and CEBP signaling. IL-6 receptor blockade impairs CD14 upregulation in infected CD34^+^ cell and reduces CFU burden, suggesting that this differentiation pathway is advantageous for *Mtb*. Interestingly, analysis of the IL6/IL6R/CEBP signaling axis among different mammalian species suggested that coordinate regulation of CEBP by IL6 and type 1 IFN is a relatively recent event in mammalian evolution and seems restricted to primates. This is corroborated by in vitro experiments using infection of cells from 10 species. Overall, the authors demonstrate, with compelling mechanistic evidence, a role for *Mtb* in co-opting an evolutionarily recent signaling axis to support infection by driving monocyte differentiation. The evolutionary aspect of the study is particularly intriguing.

Reviewer #3:

In this manuscript the authors' study the mechanisms by which HSPC differentiation occurs following *Mtb* infection. The authors uncover a less studied role for IL-6/IL-6R signaling in driving HSPC commitment to myeloid differentiation and exacerbating TB disease. In particular, re-analysis of previously published transcriptomic, proteomic, and genetic datasets support their hypotheses. Addressing the below points will further improve the impact of the findings.

Major comments:

Figure 1B-D:

The authors show that approximately 60% of CD34^+^ cells stain positively for *Mtb* with syto24, but that each of these cells has a much lower *Mtb* syto24 MFI than similarly infected CD14^+^ cells. However, after 4h, both CD34^+^ and CD14^+^ cells have a very similar number of CFU/ml.

Is the rate of uptake/infectivity potentially different when the cells are isolated and infected in a homogeneous culture rather than infected as total PBMCs?

It would also be useful to see the uninfected control quantified and included in the flow analysis as well.

Figure 3:

The authors show that heat-killed *Mtb* is unable to stimulate the same increase in myeloid differentiation of CD34^+^ cells as live *Mtb*. However, they later show that it is a cytokine (IL-6) dependent phenomenon. Are the IL-6 levels induced by heat-killed *Mtb* significantly lower than live infection?

Figure 4:

Cytokine signaling is a known trigger for myeloid cell differentiation, and therefore the authors identify differential expression of IL-6 as a potential driver of the increased number of CD14^+^ cells induced by *Mtb* infection. However, despite seeing what appears to be a significant transcriptional increase of IL-6 at 5dpi, there is no correlating increase in protein at that time point. CD34^+^ cells infected with *Mtb* and treated with aIL6R for 10d develop a lower percentage of CD14^+^ cells than untreated infected cells, at a much later time point than when differences in IL-6 induction (either by protein or transcriptional data) are seen.

A time course showing IL-6 levels in culture (1,3,5,7,10dpi) as well as a time course showing when the CD14^+^ population begins to expand in the untreated *Mtb* infected cells would more support the idea that it is an IL-6-dependent expansion. Alternatively, the authors could consider only blocking IL-6R early, when there is a significant difference in protein levels, to determine if there is the same effect on myeloid expansion.

Additionally, despite making note that "interferon signaling" and "interferon alpha/beta" pathways were significantly enriched in the infecting cells, neither IFNa/b nor IFN-g are well represented in the RNAseq data that is shown, and these cytokines are also not measured at the protein level. Including this data would provide better support for the data represented in Figure 4D-E, where no changes are seen after IFNAR or IFN-g blockade. This is a particularly critical point for the authors' later argument that transcription factors induced by both IFN and IL-6 are primarily being driven by IL-6 to induce myeloid differentiation.

The authors show in Figure 1F-G that *Mtb* is associated with cells undergoing morphological change. If exogenous IL-6 alone is sufficient to drive this myeloid differentiation, why would the cells actively infected with *Mtb* be the only cells showing signs of differentiation? The authors may want to consider additional discussion of this point.

Reviewer #4:

This manuscript investigates relationships (in human cells) between tuberculosis infection, hematopoiesis and monocyte expansion, and IL-6 signaling. In many respects the authors are investigating an aspect of 'trained' immunity or myeloid lineage 'remodeling' in the bone marrow. There are many substantial problems with this manuscript; the main one being an emphasis on correlation rather than causation. The authors attempt to draw conclusions that are predominantly indirect in nature.

1) That CD34^+^ cells can be infected in vitro does not mean it happens in vivo. The authors draw from other studies that argue that *M.tb* can get to the bone marrow. However, overall, there is no evidence that CD34^+^ cells are infected in vivo and if this would make any difference to myeloid output. The authors ignore older studies in this area (Goodell's *M. avium* Nature paper, Murray et al. Blood on the IFN-gamma KO mice infected with BCG).

2) Results section. The authors draw a conclusion "These results suggest M.tb hijacks IL-6R-mediated myeloid differentiation by human CD34^+^ cells in vivo". There is no evidence of "hijacking" provided. Only that something happens to the IL-6 pathway, which is entirely expected if a cell is infected with an intracellular bacteria.

3) Subsection “Recent genetic changes link IL6/IL6R/CEBP axis, monocyte expansion and TB pathogenesis in humans”. Myeloid expansion. In this very long section, the authors propose a correlation between myeloid expansion and M.tb infection, including a lengthy analysis of Neanderthal genomics, which cannot be tested.

In summary, the authors' model may or may not happen. The reliance on correlative studies means no firm conclusions can be drawn about the system at hand. While some leeway is warranted because it is a human-based study, the overall conclusions are not sufficiently substantiated.

---

## [Author Response]

[Editors’ note: the author responses to the first round of peer review follow.]

Our decision has been reached after consultation between the reviewers. Based on these discussions and the individual reviews below, we regret to inform you that your work will not be considered further for publication in eLife.Reviewers recognized that your manuscript provides interesting evidence illustrating that Mycobacterium tuberculosis is able to drive monocyte differentiation from human hematopoietic stem cells through the IL-6/CEBP axis. The evolutionary description of this process is also interesting. However, there are numerous problems with some of the data, both conceptual and technical, which were identified by the reviewers. Further, the establishment of causality is somewhat lacking.

We are pleased the reviewers as well as the editors found our work to be of interest and are grateful for their constructive comments for improving the manuscript. As the editors can appreciate, we have made abundant changes in the manuscript guided by the reviewers’ points and requests. Specifically, we have removed several panels and included sub-sections related to the integrated systems biology analysis in order to simplify the message.

Additionally, to remove the word “hijacking” as instructed by reviewer #4, we have now changed the title of the manuscript to “Anevolutionary recent IFN-IL6-CEBPB axis is linked to monocyte expansion and tuberculosis severity in humans”.

Finally, while the in vivo evidence presented in the manuscript is indeed based upon association and correlation, rather than causality, as is inherent to human studies and evolutionary experiments (Karmon and Pilpel, 2016), we have now presented novel data containing mechanistic insights to underscore our research hypothesis, as instructed by reviewer #4.

Reviewer #1:Background: M. tuberculosis has been detected in the bone marrow – suggestive of a possible niche for tubercle bacteria in active and latent TB. This observation also points towards a role for mycobacteria in driving differentiation of certain cell types from Hematopoietic stem cells (HSCs).Key findings include:1) The demonstration of the ability of tubercle bacteria to replicate in Hematopoietic stem cell progenitors (HSCPs), in this case CD14^+^ and CD34^+^ cells.2) Transcriptional analysis of infected CD13+ cells indicated that M. tuberculosis infection drives these cells towards myeloid differentiation, further confirmed by flow cytometry. This observation was confirmed with cells from cord blood also.3) A role for IL6R was confirmed in this myeloid differentiation pathway, using a neutralizing antibody.4) By interrogating transcriptome data, the authors confirm that the IL6R-CEBP pathway is central in monocyte expansion.5) Comparative genomics and evolutionary analysis reveal that this pathway has been the subject of recent selection.

We thank the reviewer for his/her analysis of our manuscript and for the positive comments on our manuscript.

Major Concerns:1) Figure 1—figure supplement 1C is meant to indicate that low numbers of bacteria associate with CD34^+^ cells. However, there is no comparison with CD14^+^, no quantification no statistics. This figure does not have use as it stands. Similarly, with Figure 1—figure supplement 1D – one assumes the control here would be uninfected cells? In this case, the statistical comparison shown is meaningless – comparing the bacterial load in CD14^+^ cells versus CD34^+^ cells on one graph would be more useful. The authors indicate they are comparable – but need to provide the data in a way that allows for this comparison.

We apologize for the unclear information provided. The set of experiments to study *Mtb* infectivity by CD34^+^ cells employed two approaches. First (Figure 1A-D), we used PBMC cultures, which contain both CD14^+^ and CD34^+^ cells in the same well. Second (Figure 1E-G and Figure 1—figure supplement 1C,D), we used sorted cord blood derived purified CD34^+^ cells and PBMC derived purified CD14^+^ cells from different subjects. Hence, employing confocal microscopy, now Figure 1—figure supplement 1C was meant to formally demonstrate that purified CD34^+^ cells can be infected by *Mtb*. In contrast, novel Figure 1—figure supplement 1D was meant to be a positive control confirming that H37RV *Mtb* replicates in purified CD14^+^ cells and, in that case, the statistical comparison was performed to show that the bacilli replicate over time. We apologize for the unclear information provided and have now clarified this information in the Results section and included a representative dot plot of the CD14^+^ cell population in Figure 1A.

Additionally, while we observed a statistical difference between *Mtb* replication in CD34^+^ vs CD14^+^ cells, we have not included this information in the same graph because these cells were not obtained “side-by-side” from the same donors and these cell populations were cultivated in different media – purified CD34^+^ cells require complete StemSpan SFEM II (growth factors enriched media, please see Materials and methods). Nevertheless, for comparisons and as per reviewer’s request, we have now presented this new information below (Author response image 1, left panel). As expected, we have observed that CD14^+^ cells, which are known to be highly phagocytic cells, are permissive to *Mtb* H37Rv infection. Importantly, purified CD34^+^ cells sustain H37Rv proliferation in vitro.

**Author response image 1. respfig1:** Comparative Mtb growth in purified CD14+ vs CD34+ cells. Left panel, *Mtb* growth curve in sorted purified CD34^+^ or CD14^+^ cell cultures. Right panel, *Mtb* growth curve of sorted purified CD14^+^ cells cultivated in RPMI vs StemSpan SFEM II media.

As a control (for the reviewer’s benefit), we observed that H37Rv *Mtb* growth is similar in CD14^+^ cells cultivated in complete StemSpan SFEM II media or regular RPMI media (Author response image 1, right panel) suggesting that this commercial media does not interfere with *Mtb* replication, and thus the *Mtb* growth curve as shown in Author response image 1 (left panel) is a consequence of the intracellular replication in each cell population. While direct comparisons between CD34^+^ vs CD14^+^ cells are difficult to approach, our data corroborate that CD34^+^ cells are permissive to *Mtb* H37Rv infection in vitro. Thus, we have now re-written this section and clarified this point in the Results section as a novel supplement to figure 1 (Figure 1—figure supplement 1D).

2) Figures 1 F and G need quantification, followed by statistical comparison.

Thank you for the suggestion. Quantification has been included as per reviewer’s request. The data confirms the% of infected cells at day 1 and 5 p.i. as well as the higher% of cytoplasm richer cells in *Mtb* vs uninfected cell cultures. These panels have now been incorporated in new Figure 1 and in the Results section.

3) Figure 2—figure supplement 1B and C needs statistics, are the differences significant, if not the text needs to be revised to indicate this. This applies to all supplementary graphs, these should have statistics. If differences are not significant "NS" should be stated. Same with Figures in the main text.

All graphs are now supplied with statistics. “n.s” was used for all tests in which the p value was higher than 0.05. In a few cases, such as those where found an inherent biological variation of human samples, we included the exact p values. We have also included an excel table with statistical information from each graph present in the manuscript to further clarify this section (Supplementary File 4).

4) The comparative genomics/transcriptomics and evolutionary analysis is very dense, it needs to be simplified and presented in a clearer manner.

We acknowledge the reviewer’s suggestion and we have made numerous alterations in the manuscript to present the information clearer. Specifically, we have transferred a large part of the data to the supplementary section and have removed several panels from Figures 5 and 6.

However, in reply to reviewer#4 (detailed below) regarding the ‘indirect nature’ of our data, we have also added two additional panels (Figure 6C: Chipseq analysis, chromatin architecture and CD34^+^ vs CD14^+^ cell RNAseq data; Figure 6E: increased transcription of *Mtb*-induced genes in matched European vs. African descendants with documented Neanderthal introgression), which provide solid experimental support for our hypothesis.

Overall, although our integrated system biology analysis is based on a large body of data (transcriptomics, proteomics, genomics and functional assays), we have now streamlined the evolutionary analysis into a stepwise story. Following the evolutionary timeline from early mammalian emergence (>100 million years ago (mya)), over primate (>50 mya), and hominid evolution (>15 mya), recent human evolution including Neanderthal introgression (<100,000 years ago) and human pathogen adaptation (15,000-1,500 years ago), up to extant human genetic variation (large GWAS studies). We have also added the specific timelines to each relevant panel of Figure 6 and 7, so we hope the reviewer will appreciate that the data ‘flow’ is indeed more simplified and the text easier to read.

Reviewer #2:In this manuscript Delgobo et al. demonstrate a novel role for Mtb in driving monocyte differentiation from human hematopoietic stem cells via an IL-6/CEBP axis. In vitro studies show that Mtb replicates in human CD34^+^ cells, and causes upregulation of the monocyte marker CD14 as well as enrichment of genes associated with myeloid differentiation. This in vitro monocyte expansion is demonstrated to be IL-6 dependent and associated with STAT1 and CEBP signaling. IL-6 receptor blockade impairs CD14 upregulation in infected CD34^+^ cell and reduces CFU burden, suggesting that this differentiation pathway is advantageous for Mtb. Interestingly, analysis of the IL6/IL6R/CEBP signaling axis among different mammalian species suggested that coordinate regulation of CEBP by IL6 and type 1 IFN is a relatively recent event in mammalian evolution and seems restricted to primates. This is corroborated by in vitro experiments using infection of cells from 10 species. Overall, the authors demonstrate, with compelling mechanistic evidence, a role for Mtb in co-opting an evolutionarily recent signaling axis to support infection by driving monocyte differentiation. The evolutionary aspect of the study is particularly intriguing.

We appreciate the reviewer’s analysis of our manuscript and thank him for his positive comments on the manuscript.

Reviewer #3:In this manuscript the authors' study the mechanisms by which HSPC differentiation occurs following Mtb infection. The authors uncover a less studied role for IL-6/IL-6R signaling in driving HSPC commitment to myeloid differentiation and exacerbating TB disease. In particular, re-analysis of previously published transcriptomic, proteomic, and genetic datasets support their hypotheses. Addressing the below points will further improve the impact of the findings.

We thank this reviewer for his/her positive comments on our manuscript and are grateful for his/her suggestions.

Major comments:Figure 1B-D:The authors show that approximately 60% of CD34^+^ cells stain positively for Mtb with syto24, but that each of these cells has a much lower Mtb syto24 MFI than similarly infected CD14^+^ cells. However, after 4h, both CD34^+^ and CD14^+^ cells have a very similar number of CFU/ml.Is the rate of uptake/infectivity potentially different when the cells are isolated and infected in a homogeneous culture rather than infected as total PBMCs?It would also be useful to see the uninfected control quantified and included in the flow analysis as well.

This is an interesting point. While Syto24 is a fluorescent probe employed to detect the frequency of Syto24-positive CD34^+^ vs CD14^+^ PBMC populations in the same well, this technique yields only an estimate of the numbers of *Mtb* associating with the host cell. Therefore, CFU/ml was used as a reliable golden standard bacterial count. To assess the reviewer’s question, we would need to isolate CD34^+^ cells and CD14^+^ cells from PBMC, infect with *Mtb*Syto24 and compare to CD34^+^ and CD14^+^ cells sorted from *Mtb*Syto24-infected PBMCs. This would allow us to measure side-by-side the ratio of infectivity/uptake of *Mtb* by these two cell populations from the same donors. Unfortunately, we do not have access to cell sorting equipment in our BSL-3 facility and are unable to sort *Mtb*-infected cells. Nevertheless, we have compared the *Mtb*Syto24 frequency of CD34^+^ PBMC versus purified cord blood CD34^+^ cells and found similar frequencies and MFI numbers by these cells at 4h. These data support that the rate of uptake/infectivity of CD34^+^ cells is not different when isolated or infected as (a small fraction of) total PBMCs. We have now incorporated this information in the Results section (new Figure 1—figure supplement 1B) and, as per reviewer’s request, we have now included the uninfected control in the flow cytometry analysis (new Figure 1D).

Figure 3:The authors show that heat-killed Mtb is unable to stimulate the same increase in myeloid differentiation of CD34^+^ cells as live Mtb. However, they later show that it is a cytokine (IL-6) dependent phenomenon. Are the IL-6 levels induced by heat-killed Mtb significantly lower than live infection?

The reviewer raises an interesting point but we need to correct that, based on the experimental approach employed and its limitations, in our original version of the manuscript, we have stated that *Mtb* enhanced myeloid differentiation in CD34^+^ cells by an IL-6R-mediated process. Since many myeloid differentiation markers were blocked by neutralization of IL-6R signaling in both untreated and *Mtb*-exposed cell culture systems, we deduced that this was an IL-6R-‘mediated’ and not IL-6R-‘dependent’ phenomenon. Nevertheless, as per reviewer’s request, we have now measured intracellular IL-6 levels in CD34^+^ from PBMC exposed to HK *Mtb* and live *Mtb*. Interestingly, both live Mtb and HK *Mtb* stimulates production of IL-6 as measured by flow cytometric intracellular staining (Author response image 2).

**Author response image 2. respfig2:** PBMC cultures were exposed to Mtb H37Rv (MOI=3), HK Mtb or LPS (100 ng/mL) for 24h and intracellular IL-6 was detected by flow cytometry. Live CD34^+^Lin^-^ events gated as in Figure 1A were analyzed for IL-6 MFI.

Nevertheless, such increased levels of IL-6 found in HK *Mtb* appeared not to be sufficient to induce full myeloid differentiation as seen in the live infection samples. As shown in Figure 7—figure supplement 1, HK *Mtb* does not enhance known myeloid differentiation surface markers CD4, CD64 and CD38 by progenitor CD34^+^ cells as seen in cultures exposed to live infection.

This evidence suggests that unknown activities of live pathogen infection regulates myeloid differentiation involving an IL-6R-mediated process and implies cross talking of regulatory pathways between live *Mtb*, IL-6 and interferon signaling to boost myeloid differentiation of CD34^+^ cells. While live *Mtb* strongly induces ISGs, inactivated (HK) mycobacteria do not, as seen in this study for CD34^+^ cells and others for human macrophages (Novikov et al., 2011) Additionally, *CEBPB*, a major regulator of myeloid differentiation, was not induced by HK *Mtb* but it was highly induced by live *Mtb* infection and only slightly by recombinant IL-6 treatment.

We believe live *Mtb* is a very potent stimulus to enhance STAT1 downstream pathways and ISG expression, and at least part of its actions involves an IL-6-C/EBP-β feed-forward loop in vitro. Our data suggest a collaborative and, for some functions, a redundant role for IL-6 and IFNs, C/EBP-β being an important hub for myeloid differentiation. While it has been reported that IFN-induced C/EBP-β activates GATE sequences independent of STAT1 in certain cell lineages (Li, Gade, Xiao, and Kalvakolanu, 2007), the mechanisms through which endogenous IL-6, IFN-α/β and live *Mtb* interplay to induce myeloid differentiation of HSPCs will require further exploration. We have now incorporated this data as new figures (Figure 4C and Figure 4—figure supplement 1F,G) and have included these points in the Results and Discussion section.

Figure 4:Cytokine signaling is a known trigger for myeloid cell differentiation, and therefore the authors identify differential expression of IL-6 as a potential driver of the increased number of CD14^+^ cells induced by Mtb infection. However, despite seeing what appears to be a significant transcriptional increase of IL-6 at 5dpi, there is no correlating increase in protein at that time point. CD34^+^ cells infected with Mtb and treated with aIL6R for 10d develop a lower percentage of CD14^+^ cells than untreated infected cells, at a much later time point than when differences in IL-6 induction (either by protein or transcriptional data) are seen.A time course showing IL-6 levels in culture (1,3,5,7,10dpi) as well as a time course showing when the CD14^+^ population begins to expand in the untreated Mtb infected cells would more support the idea that it is an IL-6-dependent expansion. Alternatively, the authors could consider only blocking IL-6R early, when there is a significant difference in protein levels, to determine if there is the same effect on myeloid expansion.

The reviewer raises an interesting point. The RNAseq data demonstrated in the heat maps were displayed as z-score values. However, when we plotted RNAseq normalized counts data from 2 independent experiments (technical replicates from 3 donors), we can observe that *IL6* (and *IL6R*) transcript levels are higher in *Mtb*-exposed cultures than those found in uninfected cells starting at day 1.

This was confirmed by qPCR experiments (novel Figure 4—figure supplement 1B,C), which showed enhancement of *IL6* amplification at day 1 post-infection, thus correlating with the protein being produced at this time point. However, we only started to detect a measurable difference in monocyte frequencies at 5-7 dpi. This is consistent with the idea that early signals, such as those provided by IL-6/IL-6R, are important for driving CD14^+^ monocyte differentiation at later time points. In agreement with this hypothesis, it has been reported that IL-6 increases myeloid output in multipotent progenitors but not in lineage committed myeloid progenitors (Schurch, Riether, and Ochsenbein, 2014), which implies several sequential cellular or molecular processes are needed for monocyte development to be completed. As per reviewer’s request, we have performed a detailed time course to measure CD14^+^ frequencies during *Mtb* infection by CD34^+^ cells. As a control, we confirmed that *Mtb* H37Rv replicated in the cultures over time (Author response image 3). Furthermore, CD14^+^ monocytes started to appear on day 3 of culture, but only at day 7, *Mtb* shows a slight difference in terms of CD14^+^ levels. This difference becomes more evident by day 10 of culture.

**Author response image 3. respfig3:** Mtb growth and CD14+ frequencies in CD34+ cultures exposed to H37Rv in vitro. Left panel, CFU loads of human cord blood purified CD34^+^ cells infected with Mtb H37Rv. Right panel, frequency of CD14^+^ cells in uninfected vs *Mtb* infected CD34^+^ cells over time.

It has been previously reported that early signals are sufficient to drive CD34^+^ cell differentiation (Rieger, Hoppe, Smejkal, Eitelhuber, and Schroeder, 2009; Sarrazin et al., 2009). In the published model, specific transcription factors (TFs) need to be sequentially activated to stimulate cell differentiation processes. Indeed, when we employed Cell Router on 180 hematopoiesis-associated TFs found in our transcriptomics (Figure 2B), we observed that mRNA signature scores for the myeloid development (GRAN/MONO) started at day 1 post-infection, but a difference between untreated vs *Mtb*-exposed cells was detected only after day 3 of infection. Thus, our data suggest a model in which exogenous IL-6 (in the conditioned StepSpan medium) primes secretion of endogenously produced IL-6 by live *Mtb* at day 1, which sequentially activates myeloid TFs and their targets on the following days of culture, culminating with measurable CD14^+^ cells at later time points.

However, at later time points, cell cultures change by displaying increased cell heterogeneity in addition to *Mtb* replication, which makes it difficult to approach how IL-6 responses are regulated. For example, in experiments using exogenous IL-6 (20 ng/mL) and *Mtb*, we observed that simultaneous addition of both downregulated C/EBP-β protein levels (Figure 4—figure supplement 1, compare 3^rd^ and 4^th^ lanes). It appears that, while a certain level of IL-6 production is enhanced by *Mtb*, too much cytokine could have some inhibitory actions, indicating a very complex interplay between IL-6 and *Mtb* infection. We believe that, in addition to IL-6-mediated myeloid differentiation by *Mtb* initiated early on during infection, pathogen replication regulates yet-to-be-defined intracellular associated events. We appreciate the reviewer’s suggestion about the early blockade of IL-6, but unfortunately, we did not have enough cells to perform this experiment. Nevertheless, our data show that IL-6R participates in the *Mtb*-enhanced myeloid differentiation and we have now incorporated the monocyte differentiation curve data (Figure 3—figure supplement 1E) in the Results section.

Additionally, despite making note that "interferon signaling" and "interferon alpha/beta" pathways were significantly enriched in the infecting cells, neither IFNa/b nor IFN-g are well represented in the RNAseq data that is shown, and these cytokines are also not measured at the protein level. Including this data would provide better support for the data represented in Figure 4D-E, where no changes are seen after IFNAR or IFN-g blockade. This is a particularly critical point for the authors' later argument that transcription factors induced by both IFN and IL-6 are primarily being driven by IL-6 to induce myeloid differentiation.

We were also intrigued with the increased ISG expression in the RNAseq without IFN-α/β and IFN-γ being represented in the RNAseq data. However, some but not all donors displayed increased *IFN* mRNA as detected in qPCR experiments (Figure novel S4c), which could partially explain the observed "interferon signaling" and "interferon α/β" pathways enriched in our samples. However, several groups previously demonstrated that while ISG expression can be easily detected, *IFNA/B* mRNA or protein are notoriously difficult to detect (Berry et al., 2010; Rodero et al., 2017). For example, a recent study (Llibre et al., 2019) employed digital ELISA to show neither type I IFN was detected in samples from tuberculosis patients, whereas type I IFN signaling is detectable in myeloid cells after infection (Berry et al., 2010). Nevertheless, as per reviewer request, we measured IFN-γ in the supernatants by ELISA, which showed no production of this cytokine in CD34^+^ cell cultures exposed to *Mtb*. Unfortunately, we did not have access to the ultrasensitive assay used by Rodero et al. and Llibre et al. to measure type I IFNs. However, anticipating the difficult to measure type I IFN in our system, we have performed experiments blocking IFNAR2, demonstrating inhibition of *Mtb*-enhanced ISGs STAT1 and MX1, suggesting bioactive type I IFN protein is present in our system and involved in the ISGs transcription upregulated in CD34^+^ cells. We have now included this new set of data in a new Figure 4—figure supplement 1(E and H) and have mentioned this in the Results section of the manuscript.

The authors show in Figure 1 F-G that Mtb is associated with cells undergoing morphological change. If exogenous IL-6 alone is sufficient to drive this myeloid differentiation, why would the cells actively infected with Mtb be the only cells showing signs of differentiation? The authors may want to consider additional discussion of this point.

We thank the reviewer for raising this point. We also observed cytoplasm-richer cells in the uninfected cultures, albeit at lower numbers when compared to *Mtb*-exposed cells. Originally, we intended to show that *Mtb* enhances cellular differentiation when compared to uninfected cell cultures. As per request of reviewer #1, we have included quantification of “signs of differentiation” in the Results section (Figure 1F-G). We apologize for the lack of clarity and as per reviewer’s request we have now clarified this point in the Results section.

Reviewer #4:This manuscript investigates relationships (in human cells) between tuberculosis infection, hematopoiesis and monocyte expansion, and IL-6 signaling. In many respects the authors are investigating an aspect of 'trained' immunity or myeloid lineage 'remodeling' in the bone marrow. There are many substantial problems with this manuscript; the main one being an emphasis on correlation rather than causation. The authors attempt to draw conclusions that are predominantly indirect in nature.

We acknowledge this reviewer for his/her comments on our manuscript and have addressed the topic of correlation vs. causation, as well as the length of the evolutionary analysis. While we did not indicate in the paper our findings were “causal”, we have made many substantial changes in the manuscript which now provide further mechanistic insights and additional experimental supporting a role for the IFN-IL6-CEBP gene module in TB susceptibility connected to myeloid differentiation.

Though the reviewer’s comment that we are investigating of *“an aspect of trained immunity or myeloid lineage remodeling*’” could be a possibility, based on our study, we cannot conclude that the Mtb-activated IFN-IL6-CEBP gene module directly induces *de facto* innate immune memory (Netea et al., 2016) in vivo. To do so, we would need to investigate cohorts of re-infected patients and interrogate whether the IFN-IL6-CEBP axis is modulated during re-infection and linked to different TB disease outcomes. Unfortunately, to our knowledge, no such cohorts exist at the present moment and the correlates of innate immune memory during the natural infection of *Mtb* are still an open question in the field. Yet, our results argue in favor of the IFN-IL6-CEBP axis amplifying, rather than protecting against TB disease in humans. In support of this hypothesis, HSPC-derived monocytes induced by *Mtb* display increased frequency of CD14^+^CD16^+^ cells (Figure 3J), a subpopulation associated with severe TB disease (Balboa et al., 2011).

1) That CD34^+^ cells can be infected in vitro does not mean it happens in vivo. The authors draw from other studies that argue that M.tb can get to the bone marrow. However, overall, there is no evidence that CD34^+^ cells are infected in vivo and if this would make any difference to myeloid output.

We respectfully disagree with the reviewer since it has been previously demonstrated that CD34^+^ cells are indeed infected with *Mtb* in vivo in humans. We have cited these studies in the original version of the manuscript (Tornack et al., 2017). In addition, as summarized in Mert et al. (Mert et al., 2001), 39 out of 149 bone marrow biopsies in 6 case series of miliary TB patients were positive for *Mtb*, by Ziehl-Neelsen staining and/or PCR. It should be stated that the samples used in their study were either obtained at autopsy or for diagnostic purpose. Due to obvious ethical limitations, we did not have access to bone marrow samples from patients with active TB, but have compared our results obtained in CD34^+^ cells obtained from purified cord blood and PBMCs with CD34^+^ cells from healthy bone marrow, with similar results (novel Figure 2—figure supplement 1B,C). Regarding myeloid output in TB, we have also cited early studies showing that monocytes are increased in active pulmonary TB patients (Rogers, 1928; Schmitt, Meuret, and Stix, 1977). Furthermore, we have also cited more recent experiments by Scriba et al. (Scriba et al., 2017) demonstrating that monocytes are increased during activation of latent TB in vivo. Together, these studies provide a theoretical framework raising the possibility that *Mtb* infection enhances monocyte output during active TB disease in vivo. We have now stressed this point in the Introduction section of the manuscript.

The authors ignore older studies in this area (Goodell's M. avium Nature paper, Murray et al. Blood on the IFN-gamma KO mice infected with BCG).

We did cite the work by Goodell and colleagues in our previous version of the manuscript (Baldridge, King, Boles, Weksberg, and Goodell, 2010). In this mouse model study, proliferation of HSPCs was induced by IFN-γ, whereas anti-IFN-gamma neutralizing antibody had no effect on monocyte expansion in our human HSPC model. While Murray, Young and Daley (1998) found a role for IFN-g in the extramedullary expansion of myeloid cells in BCG-infected mice, we did not detect production of IFN-γ in our supernatants of *Mtb* H37Rv-exposed human primary CD34^+^ cells. It should also be noted that mice infected with *M. avium, M. tuberculosis* or *M. bovis* BCG show different outcomes of disease such as cellular dynamics of myeloid cells circulating in the blood and/or recruited into the tissues. For instance, Noris and Ernst have recently reported *Mtb*-infected mice displayed increased egress of monocytes from the bone marrow to the blood (Norris and Ernst, 2018). In contrast, Baldridge et al. observed that, upon infection with *M. avium*, the absolute numbers of lymphocytes, neutrophils, monocytes, eosinophils and basophils in peripheral blood remained stable; with a relative increase of CD4^+^ T-cells and granulocytes. Moreover, Baldridge et al. showed a loss of granulocyte-monocyte progenitors, common myeloid progenitors, and macrophage-erythroid progenitors with a concomitant increase in common lymphoid progenitors in murine bone marrow. These changes are in strong contrast with human tuberculosis data, where several groups have demonstrated lymphopenia, in parallel to myeloid expansion (both granulocytes and monocytes), as discussed above in point #1.1. We apologize for not citing previous work on murine models of BCG infection such as *Murray* et al. 1998, as our manuscript is focusing on human pathogenesis caused by *Mtb* infection. We have now cited the article by *Murray* et al. 1998 and in order to shorten and streamline the systems biology and evolutionary part of the manuscript, we have also removed data analysis of human BCG vaccination.

2) Results section. The authors draw a conclusion "These results suggest M.tb hijacks IL-6R-mediated myeloid differentiation by human CD34^+^ cells in vivo". There is no evidence of "hijacking" provided. Only that something happens to the IL-6 pathway, which is entirely expected if a cell is infected with an intracellular bacteria.

We would like to correct that our manuscript actually stated: “These results suggest *Mtb* hijacks IL-6R-mediated myeloid differentiation by human CD34^+^ cells in vitro.” We suggested the term “hijacking” because, in our opinion, this correctly and intuitively describes the in vitro process in which *Mtb* infection of purified CD34^+^ cells significantly increase monocyte expansion, resulting in significantly increased mycobacterial burden, both of which are blocked by anti-IL-6R antibody, the latter arguing in favor of causality, at least in vitro. However, as instructed by the Editor, we have now removed the word “hijacking” from the manuscript. We hypothesize this phenomenon might occur in vivo, since progression from latent to active TB is accompanied with increased monocyte levels (novel Figure 5F), significantly correlated with the IL-6 signaling pathway (novel Figure 5H-G-I, p<0.0001), as well as in vivo *Mtb* loads in TB patients (novel Figure 5J*CEBPB* transcript levels versus *Mtb* sputum positivity, p<0.01), all of this in paired samples (also expanded with disease severity measured as symptom count and modal X-ray grade in Suppl. Figure S5b). This in vivo evidence is indeed based upon association/correlation and we did not indicate causation in any of this data interpretation.

Weagree with the reviewer that IL-6 (among other pro-inflammatory cytokines) is frequently induced by upon infection by several bacteria. However, this does not translate into increased monocyte expansion in several murine models of infection. As outlined above, Baldridge et al. (2010) demonstrated strictly IFN-γ-dependent, IFNAR-independent HSPC proliferation upon *M. avium* infection, without any effect on circulating monocytes. On the other hand, MacNamara et al. described IFN-γ-dependent, IFNAR-independent monocytosis upon *Ehrlichia muris* infection (MacNamara et al., 2011). Boettcher and colleagues found G-CSF-dependent granulopoiesis upon LPS exposure and *E. coli* infection (Boettcher et al., 2014), while Granick et al. show PGE2-dependent granulopoiesis upon *S. aureus* infection (Granick et al., 2013). Since these models do not reflect physiological infection of the bone marrow, we have also compared *Mtb* with *Leishmania infantum*, which is a classical human bone marrow pathogen and the causative agent of human visceral leishmaniasis. As outlined above in reply to Reviewer#2, we found that *L. infantum* infection of human CD34^+^ cells does not increase monocyte output in vitro, in contrast to *Mtb* infection. We have now clarified this point in the Discussion section.

3) Subsection “Recent genetic changes link IL6/IL6R/CEBP axis, monocyte expansion and TB pathogenesis in humans”. Myeloid expansion. In this very long section, the authors propose a correlation between myeloid expansion and M.tb infection, including a lengthy analysis of Neanderthal genomics, which cannot be tested.

As outlined in the reply to reviewer#1, we have now streamlined the evolutionary analysis into a stepwise story. Following the evolutionary timeline from early mammalian emergence (>100 million years ago (mya)), over primate (>50 mya), and hominid evolution (>15 mya), recent human evolution including Neanderthal introgression (<100,000 years ago) and human pathogen adaptation (15,000-1,500 years ago), up to extant human genetic variation (large GWAS studies). We have also added the specific timelines to each relevant panel of Figure 6, so we hope the reviewer will appreciate that the data ‘flow’ is indeed more simplified and the text easier to read. In addition, we have removed a large part of data analysis from the text and also removed several panels from Figure 5 and 6.

Nevertheless, regarding “…*Neanderthal genomics, which cannot be tested*”, we respectfully disagree with the reviewer. We have tested our evolutionary hypothesis on several independent levels, using a data-driven approach by integrating our findings in CD34^+^ cells with publicly available transcriptomic, genetic and functional genomic data generated by the scientific community. Starting with several datasets spanning mammalian evolution, we generated independent experimental confirmations providing strong evidence in favor of our observations: 1. amino acid conservation levels in IL6/IL6R/CEBPA/CEBPB/CEBPD*, 2.* human- and primate-specific IFN regulation of *CEBPB* and *CEBPD* transcripts as compared to other mammals or rodents (novel Figure 6A, Figure 6B-C and Figure 5—figure supplement 1D).

Regarding human genetics/evolution, of which Neandertal introgression analysis is a small (but significant) part (see below, novel Figure 6D), GWAS data validated in several independent cohorts from >230,000 individuals (>170,000 Europeans, >62,000 Japanese) demonstrate an irrefutable genetic link between systemic monocyte levels and polymorphisms in several genes from our proposed IL6/IL6R/CEBP gene module (summarized in Figure 7A and detailed in Supplementary file 2), of which a significant number are also validated TB susceptibility genes (Suppl. Figure 5f). In the absence human “knock-out” models, GWAS of large case-control studies are the closest equivalent, several of which have been able to use Mendelian randomization to investigate causation (Porcu et al., 2019; Taylor et al., 2019; Warrington et al., 2019).

In summary, the authors' model may or may not happen. The reliance on correlative studies means no firm conclusions can be drawn about the system at hand. While some leeway is warranted because it is a human-based study, the overall conclusions are not sufficiently substantiated.

We respectfully disagree with the reviewer that we did not present substantiated evidence to propose this model (which has not been challenged by the other 3 reviewers). Nevertheless, we acknowledge his/her comment and have now included two new Figure panels, which provide mechanistic insights and additional experimental support for our model.

First, to substantiate our hypothesis that transcriptional up-regulation of *CEBPB* and *CEBPD* by IFN are recent in mammalian evolution, we compiled IFN signaling through STAT1 ChipSeq analysis with chromatin architecture, RNAseq gene transcription in purified CD34^+^ vs. CD14^+^ cells and sequence conservation across 100 vertebrates (ranging from birds to mammals). We compared the most diverse genes in our module, *CEBPB* and *CEBPD,* to the most conserved, *CXCL10* and *CXCL9*, with respect to their type I IFN regulation (represented in Figure 6B). As shown in Figure 6C, we found that STAT1-binding peaks in *CEBPB* and *CEBPD* observed by ChipSeq analysis of IFN-stimulated human monocytes corresponded to regions with active chromatin (DNase Hypersensitivity Sites, DHS) and correlated with increased downstream transcription in CD14^+^ monocytes and purified CD34^+^ cells. In agreement with our findings (represented in Figure 6B), only 3 out of 11 (27%) STAT1 binding peaks in *CEBPB* and *CEBPD* were found in regions conserved during mammalian evolution (from chicken to humans), while 6 out of 7 (86%) STAT1 peaks were conserved in *CXCL9* and *CXCL10* genes.